# T-705-Derived Prodrugs Show High Antiviral Efficacies against a Broad Range of Influenza A Viruses with Synergistic Effects When Combined with Oseltamivir

**DOI:** 10.3390/pharmaceutics15061732

**Published:** 2023-06-14

**Authors:** Benedikt Ganter, Martin Zickler, Johanna Huchting, Matthias Winkler, Anna Lüttjohann, Chris Meier, Gülsah Gabriel, Sebastian Beck

**Affiliations:** 1Organic Chemistry, Department of Chemistry, Faculty of Sciences, Hamburg University, 20146 Hamburg, Germany; benedikt.ganter@uni-hamburg.de (B.G.); johanna.huchting@itmp.fraunhofer.de (J.H.); matthias.winkler@uni-hamburg.de (M.W.); 2Department for Viral Zoonoses-One Health, Leibniz Institute of Virology, 20251 Hamburg, Germany; martin.zickler@outlook.de (M.Z.); anna.luettjohann@leibniz-liv.de (A.L.); sebastian.beck@leibniz-liv.de (S.B.); 3Fraunhofer Institute for Translational Medicine and Pharmacology ITMP, 22525 Hamburg, Germany; 4German Center for Infection Research (DZIF), 38124 Braunschweig, Germany; 5Institute of Virology, University of Veterinary Medicine Hannover, 30559 Hannover, Germany

**Keywords:** influenza A virus, antiviral drug, Favipiravir, T-705, T-1105, T-1106, nucleotide prodrugs of T-1106, drug synergy, pandemic preparedness

## Abstract

Emerging influenza A viruses (IAV) bear the potential to cause pandemics with unpredictable consequences for global human health. In particular, the WHO has declared avian H5 and H7 subtypes as high-risk candidates, and continuous surveillance of these viruses as well as the development of novel, broadly acting antivirals, are key for pandemic preparedness. In this study, we sought to design T-705 (Favipiravir) related inhibitors that target the RNA-dependent RNA polymerase and evaluate their antiviral efficacies against a broad range of IAVs. Therefore, we synthesized a library of derivatives of T-705 ribonucleoside analogues (called T-1106 pronucleotides) and tested their ability to inhibit both seasonal and highly pathogenic avian influenza viruses in vitro. We further showed that diphosphate (DP) prodrugs of T-1106 are potent inhibitors of H1N1, H3N2, H5N1, and H7N9 IAV replication. Importantly, in comparison to T-705, these DP derivatives achieved 5- to 10-fold higher antiviral activity and were non-cytotoxic at the therapeutically active concentrations. Moreover, our lead DP prodrug candidate showed drug synergy with the neuraminidase inhibitor oseltamivir, thus opening up another avenue for combinational antiviral therapy against IAV infections. Our findings may serve as a basis for further pre-clinical development of T-1106 prodrugs as an effective countermeasure against emerging IAVs with pandemic potential.

## 1. Introduction

Influenza A viruses (IAV) remain one of the major causative viral agents of acute respiratory illness and associated deaths worldwide. IAV can give rise to annual epidemics and occasional pandemics. According to estimations by the WHO, seasonal influenza epidemics may cause approximately 1 billion symptomatic infections worldwide, including 3–5 million severe cases and 290,000–600,000 influenza-related respiratory deaths every year [1]. Upon animal-to-man transmission, new IAV strains might emerge to which no or only limited immunity in a naïve human population exists. Further adaptation of these viruses to humans, allowing sustained human-to-human transmission, can lead to a pandemic of unpredictable magnitude and consequences for global health. The ongoing COVID-19 pandemic, likely caused by the transmission of a novel coronavirus from an animal reservoir [2], as well as sporadic spillovers of avian influenza viruses to mammalian species, including humans, most recently seen by H5N1 highly pathogenic avian influenza viruses (HPAIV) [3], further highlight the ever-present pandemic threat of novel (respiratory) viruses emerging from the animal kingdom.

Pandemic preparedness is key for rapid response measures if a novel IAV crosses the species barrier and becomes an air-borne pandemic virus. Herein, both vaccines and therapeutics are of great importance in reducing the number of hospitalizations and deaths, as well as in preventing the breakdown of worldwide healthcare systems and economies. However, developing new vaccines against the pandemic agent takes time, and thus only existing therapeutics present an option for a rapid first response until a vaccine becomes available. Unfortunately, so far, the available therapeutic options to treat influenza infections are limited [4]. For both M2 ion channel inhibitors (adamantanes) and neuraminidase inhibitors such as oseltamivir (Tamiflu), resistance-conferring mutations have been repeatedly observed [5]. Moreover, even though oseltamivir has been successfully used to treat rather mild influenza cases over the past two decades, its effectiveness in preventing severe influenza complications is still unclear [6].

More recently, compounds targeting the influenza virus polymerase complex (consisting of subunits PB1, PB2, and PA) have gained attention [7]. For example, clinical success was demonstrated for baloxavir (Xofluza), an inhibitor of the endonuclease activity in the PA-subunit, in patients with uncomplicated influenza A and B, leading to FDA approval in October 2018 [8]. However, shortly after being licensed for widespread clinical use, several studies reported low therapeutic efficacy of baloxavir due to a high mutation rate, particularly in children [9]. Pimodivir, an inhibitor targeting the cap-binding function of the PB2 protein, was developed by Janssen Pharmaceuticals; however, further clinical development was discontinued in 2021 due to reduced efficacy over standard or care therapy [10]. To date, by screening large small-molecule libraries or employing structure-based drug design, numerous other studies reported on the identification of novel compounds that specifically target the influenza RNA-dependent RNA polymerase (RdRP) through multiple mechanisms [11,12,13,14,15,16]. In addition to broad-spectrum antiviral activity, a subset of these compounds also showed potent synergistic activity with neuraminidase inhibitors, such as zanamivir [11]. Novel, innovative approaches further focus on specifically targeting viral proteins for degradation using the PROTAC (proteolysis targeting chimera) system [17]. For example, PROTACs have been successfully used for targeted degradation of the viral surface glycoproteins, hemagglutinin and neuraminidase, and this system might be easily adaptable to the RdRP in the future [18,19]. Collectively, these data highlight the viral polymerase complex as a promising target for therapeutic intervention; however, since currently licensed RdRP inhibitors are often associated with low-to-medium clinical efficacy and the emergence of drug-resistant variants, there is an urgent demand for novel (RdRP-directed) antivirals.

T-705 (alias Favipiravir, Avigan), a nucleobase analogue, was licensed in 2014 in Japan to treat emerging and re-emerging influenza viruses that show resistance to existing medications [20]. In contrast to Baloxavir, which specifically inhibits the endonuclease activity of the PA polymerase subunit, the main mode of action for T-705 was shown to be lethal mutagenesis [21]. After intracellular conversion into the corresponding ribonucleoside analogue and the 5′-triphosphate thereof (T-705-TP), the IAV polymerase recognizes T-705-TP as a substrate alternative to ATP/GTP and incorporates it during RNA genome replication [22]. Ultimately, this leads to a massive accumulation of mutations and, as a consequence, the extinction of infectious virus particles. In general, the T-705 mode of action is regarded as highly attractive due to its broad-range activity against several RNA viruses [23,24]. However, the enthusiasm is hampered by reports that T-705 might have teratogenic and embryotoxic effects, which prevent its use in pregnant women, who are at the highest risk of developing severe influenza [25]. Moreover, some studies show that T-705 has a rather low efficacy in alleviating symptoms in patients with uncomplicated influenza, even at high treatment doses, most likely due to inefficient intracellular conversion of the nucleobase to the active metabolite [26]. Finally, T-705 is chemically unstable when it is metabolized into its nucleosidic/nucleotidic form, which negatively affects its antiviral activity [27].

Due to the high chemical instability of any ribosylated and phosphoribosylated forms of T-705, a de-fluoro analogue (termed T-1105) was recently described [27]. However, both T-705 and T-1105 are inefficiently metabolized into their antiviral active triphosphate forms by human enzymes [28]. In order to bypass this metabolic bottleneck in T-705/T-1105 activation, a nucleosidic form of T-1105, called T-1106, was synthesized that, after conversion into the corresponding triphosphate thereof (T-1106-TP), showed increased antiviral activity and chemical stability compared to T-705-TP [29]. The mechanism of inhibition of the influenza polymerase has recently been described using cryo-EM structure analysis [30].

In this study, we aimed to further improve T-1106 function by preparing various T-1106-derived nucleosides as well as delivering the nucleoside-5′-di-(T-1106-DP) and triphosphates (T-1106-TP) directly into the cell. To enable the polar di- and triphosphates to pass the cell membrane, the Di- and Tri*PPP*ro-prodrug concept that was recently developed by us was applied [31]. We demonstrate that particularly T-1106-DP pronucleotides are highly active against IAV of both human and avian origin in vitro and that they act synergistically with the neuraminidase inhibitor oseltamivir against influenza viruses, such as H5N1 HPAIV, with pandemic potential.

## 2. Materials and Methods

### 2.1. Cell Lines

Madin-Darby canine kidney (MDCKII) cells (ATCC CRL-2936) were cultivated in Minimal Essential Medium (MEM) supplemented with 10% fetal bovine serum (FBS), 1% l-glutamine and 1% Penicillin/Streptomycin (Sigma-Aldrich, St. Louis, MO, USA). Cells were incubated under humidified conditions at 37 °C and 5% CO_2_ and regularly verified to be mycoplasma-negative.

### 2.2. Viruses

Influenza viruses used in this study included A/Hamburg/NY1580/09 (H1N1) [32], A/Victoria/03/75 (H3N2) [33], A/Vietnam/1194/04 (H5N1) (kind gift from Hans-Dieter Klenk, University of Marburg, Marburg, Germany) and A/Anhui/1/13 (H7N9) [34]. All experiments using H5N1 and H7N9 HPAIV were conducted in a laboratory approved for biosafety level 3 (BSL-3) work at the Leibniz Institute of Virology, Hamburg, Germany.

### 2.3. Virus Propagation

All influenza A virus subtypes used in this study were propagated on MDCKII cells in infection medium (MEM supplemented with 1% Penicillin/Streptomycin, 0.2% bovine serum albumin (BSA; Sigma-Aldrich, St. Louis, MO, USA) and 1 µg/mL TPCK-treated trypsin (Sigma-Aldrich, St. Louis, MO, USA)). At approximately 36 to 48 h post-inoculation, the cell culture supernatant was harvested and centrifuged at 2000× *g* and 4 °C for 10 min. The cleared supernatant was passed through a 0.45 µm syringe filter, aliquoted, and stored at −80 °C. Infectious virus titers were determined by plaque assay.

### 2.4. Virus Infection

MDCKII cells were seeded in 24-well tissue culture plates with 2.5 × 10^5^ cells per well one day prior to infection. The virus inoculum with a defined multiplicity of infection (MOI) of 0.01 was prepared in an inoculation medium (MEM, 1% Penicillin/Streptomycin, 0.2% BSA). Cells were washed once with sterile phosphate-buffered saline (PBS; Sigma-Aldrich, St. Louis, MO, USA) and infected for 30 min at 37 °C, followed by two additional PBS washing steps. Subsequently, 1 mL of infection medium (MEM, 1% Penicillin/Streptomycin, 0.2% BSA, and 1 µg/mL TPCK-treated trypsin), supplemented with or without the test compounds, was added to each well, and the cells were further incubated for 24 h at 37 °C. At 24 h post-infection, cell culture supernatants were collected, and infectious virus titers were determined by plaque assay as described below.

### 2.5. Inhibitors

T-705 was purchased from ChemBlock (Hayward, CA, USA), and T-1105 from Accela ChemBio Inc. (San Diego, CA, USA). The nucleosides **2a**, **2b**, and **2c** were synthesized based on previously reported protocols [29]. The derivatives **3a** and **3b** were synthesized according to published protocols at the chemistry department of the University of Hamburg, Hamburg, Germany [29]. Chemical synthesis pathways for those compounds, which are described for the first time in the present study, are described in detail below.



**Synthesis of Methyl-3-Hydroxy-2-pyrazinecarboxylate (7)**


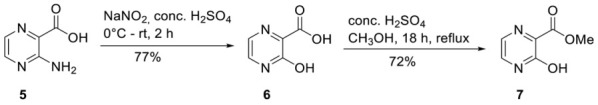



**3-Hydroxypyrazine-2-carboxylic acid 6.** 3-Amino-2-pyrazinecarboxylic acid **5** (20.0 g, 144 mmol) was suspended in 100 mL conc. H_2_SO_4_. At 0 °C, sodium nitrite (14.9 g, 216 mmol) was added portion-wise. The suspension was afterward stirred at room temperature for two hours until a clear solution was formed. The reaction was quenched by purring the reaction mixture in ice water (500 mL). The precipitated solid was collected by vacuum filtration and washed with ice-cold water, and redissolved in a 5%-sodium carbonate solution. 3-Hydroxypyrazine-2-carboxylic acid **6** was precipitated by the addition of 1 M HCl (300 mL), collected by vacuum filtration, washed with ice cold water, and dried under vacuum to give 16.0 g of **6** (114 mmol, 77%) as a brownish solid and was used without further purification. ^1^H-NMR (300 MHz, DMSO-*d*_6_): δ [ppm] = 7.80 (d, ^3^*J*_H,H_ = 3.7 Hz, 1H, H-5), 7.65 (d, ^3^*J*_H,H_ = 3.7 Hz, 1H, H-6). **Methyl-3-hydroxy-2-pyrazinecarboxylate 7. 6** (16.0 g, 114 mmol) was dissolved in MeOH (300 mL) and conc. H_2_SO_4_ (2 mL) was added. After the reaction mixture was refluxed overnight all volatiles were removed under reduced pressure. Purification on silica gel (CH_2_Cl_2_/MeOH; 30:1) provided 14.0 g of **7** (90.8 mmol, 72%) as a yellow solid. ^1^H-NMR (300 MHz, DMSO-*d*_6_): δ [ppm] = 7.71 (d, ^3^*J*_H,H_ = 3.7 Hz, 1H, H-5), 7.47 (d, ^3^*J*_H,H_ = 3.7 Hz, 1H, H-6), 3.81 (s, 3H, C*H*_3_-Ester).



**Synthesis of 5-*O*-Benzoyl-1,2-di-*O*-acetyl-3-deoxy-D-ribofuranose (13)**


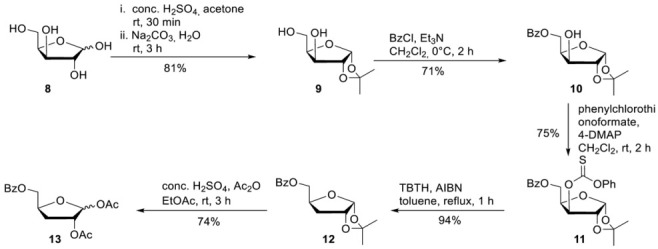



**1,2-*O*-Isopropylidene-alpha-D-xylofuranose 9.** D-Xylose **8** (20.0 g, 0.133 mmol) was dissolved in acetone (520 mL) after the addition of conc. H_2_SO_4_ (20 mL), the reaction mixture was stirred for 30 min. A solution of Na_2_CO_3_ (26.0 g, 250 mmol) in water (224 mL) was added while the reaction mixture was cooled in an ice bath. After the reaction mixture was stirred for a further 3 h at room temperature, solid Na_2_CO_3_ was added until pH = 7. The precipitated solid was collected by vacuum filtration and washed with acetone. Purification of the crude product on silica gel (CH_2_Cl_2_/MeOH; 50:1) provided 20.5 g of **9** (107 mmol, 81%) as a colourless solid. ^1^H-NMR (300 MHz, DMSO-*d*_6_): δ [ppm] = 5.80 (d, ^3^*J*_HH_ = 3.9 Hz, 1H, H-1), 5.13 (d, ^3^*J*_HH_ = 4.9 Hz, 1H, 3-O*H*), 4.61 (t, ^3^*J*_HH_ = 5.6 Hz, 1H, 5-O*H*), 4.36 (d, ^3^*J*_HH_ = 3.8 Hz, 1H, H-3), 4.00–3.94 (m, 2H, H-2, H-4), 3.64–3.46 (m, 2H, H-5), 1.37 (s, 1H, C-CH_3_), 1.22 (s, 1H, C-CH_3_). **5-*O*-Benzoyl-1,2-*O*-isopropylidene-alpha-D-xylofuranose 10.** To a solution of **9** (20.5 g, 107 mmol) in CH_2_Cl_2_ (440 mL) was added Et_3_N (22.4 mL g, 161 mmol). Subsequently, benzoyl chloride (13.6 mL, 117 mmol) was added dropwise at 0 °C for 10 min. After the reaction mixture was stirred 2 h at 0 °C, the reaction was quenched by the addition of saturated NaHCO_3_. The organic phase was washed with saturated NaHCO_3_ and dried over Na_2_SO_4_. All volatiles were removed under reduced pressure, and purification of the crude product on silica gel (PE/EE; 2:1) provided 22.5 g of **10** (76.4 mmol, 71%) as a colourless solid. ^1^H-NMR (300 MHz, DMSO-*d*_6_): δ [ppm] = 8.00–7.93 (m, 2H, H-Bz), 7.71–7.64 (m, 1H, H-Bz), 7.56–7.51 (m, 2H, H-Bz), 5.90 (d, ^3^*J*_H,H_ = 3.7 Hz, 1H, H-1), 5.49 (d, ^3^*J*_H,H_ = 4.9 Hz, 1H, 3-OH), 4.51–4.31 (m, 3H, H-3, H-4, H-5), 4.13 (dd, ^3^*J*_H,H_ = 5.0 Hz, ^4^*J*_H,H_ = 2.5 Hz, 1H, H-2), 1.40 (s, 1H, C-CH3), 1.25 (s, 1H, C-CH3). **5-*O*-Benzoyl-3-*O*-phenylcarbonothioat-1,2-*O*-isopropylidene-alpha-D-xylofuranose 11.** To a solution of **10** (318 mg, 1.08 mmol) in dry CH_2_Cl_2_ (10 mL) was added 4-DMAP (330 mg, 2.70 mmol) and phenylchlorothionoformate (180 µL, 1.30 mmol). After the reaction mixture was stirred for 2 h at room temperature, it was washed subsequently with a solution of 0.5 M HCl, 0.5 M NaOH, and saturated NaCl solution and dried over Na_2_SO_4_. All volatiles were removed under reduced pressure, and purification of the crude product on silica gel (PE/EA 5:1) provided 348 mg of **11** (0.81 mmol, 75%) as a colourless solid. ^1^H-NMR (300 MHz, DMSO-*d*_6_): δ [ppm] = 8.02–7.99 (m, 2H, H-Bz), 7.71–7.65 (m, 1H, H-Bz), 7.57i–7.45 (m, 4H, H-Bz, H-Phenyl), 7.36–7.33 (m, 1H, H-Phenyl), 7.16–7.11 (m, 1H, H-Phenyl), 6.06 (d, ^3^*J*_H,H_ = 3.9 Hz, 1H, H-1), 5.68 (d, ^3^*J*_H,H_ = 2.9 Hz, 1H, H-3), 4.91 (d, ^3^*J*_H,H_ = 4.0 Hz, 1H, H-2) 4.73–4.68 (m, 1H, H-4), 4.60 (dd, ^2^*J*_HH_ = 11.5 Hz, ^3^*J*_HH_ = 4.7 Hz, 1H, H-5), 4.48 (dd, ^2^*J*_HH_ = 11.5 Hz, ^3^*J*_HH_ = 4.7 Hz, 1H, H-5), 1.48 (s, 1H, C-CH3), 1.30f (s, 1H, C-CH3). **5-*O*-Benzoyl-3-deoxy-1,2-*O*-isopropylidene-alpha-D-xylofuranose 12.** To a solution of **11** (340 mg, 0.79 mmol) in dry toluene (10 mL) was added AIBN (26 mg, 0.16 mmol) and tributyltin hydride (0.29 mL, 1.16 mmol). Afterward, the reaction flask was put into a on 110 °C preheated oil bath. After the reaction mixture was stirred for 2 h at 110 °C all volatiles were removed under reduced pressure, and purification of the crude product on silica gel (PE/EA 4:1) provided 208 mg of **12** (0.74 mmol, 94%) as a colourless oil. ^1^H-NMR (300 MHz, DMSO-*d*_6_): δ [ppm] = 8.00–7.96 (m, 2H, H-Bz), 7.70–7.64 (m, 1H, H-Bz), 7.57–7.51 (m, 2H, H-Bz), 5.80 (d, ^3^*J*_H,H_ = 3.7 Hz, 1H, H-1), 4.78 (t, ^3^*J*_H,H_ = 4.3 Hz, 1H, H-2), 4.47 (dd, ^2^*J*_H,H_ = 11.4 Hz, ^3^*J*_H,H_ = 3.0 Hz, 1H, H-5), 4.43–4.37 (m, 1H, H-4), 4.29 (dd, ^2^*J*_H,H_ = 11.6 Hz, ^3^*J*_H,H_ = 5.3 Hz, 1H, H-5), 2.08–2.03 (m, 1H, H-3), 1.80–1.70 (m, 1H, H-3), 1.41 (s, 1H, C-CH3), 1.25 (s, 1H, C-CH3). **1,2-*O*-Diacetyl-5-*O*-benzoyl-3-deoxy-D-xylofuranose 13. 12** (2.90 g, 10.4 mmol) was added to a solution of conc. H_2_SO_4_ (1.5 mL), acetic acid (80 mL), and acetic anhydride (8.0 mL) under cooling. After the reaction mixture was stirred for 6 h, ice water (150 mL) was added, and the mixture was extracted with CH_2_Cl_2_. The organic phase was washed with saturated Na_2_SO_3_ and NaCl solution and dried over Na_2_SO_4_. Afterwards all volatiles were removed under reduced pressure, and purification of the crude product on silica gel (PE/EA 5:1) provided 2.49 g of **12** (7.73 mmol, 74%) as a colourless oil. ^1^H-NMR (300 MHz, CDCl_3_): δ [ppm] = 8.11–8.05 (m, 2H, H-Bz), 7.62–7.57 (m, 1H, H-Bz), 7.50–7.44 (m, 2H, H-Bz), 6.22 (s, 1H, H-1), 5.26–5.25 (m, 1H, H-2), 4.78–4.70 (m, 1H, H-4) 4.54 (dd, ^2^*J*_H,H_ = 12.0 Hz, ^3^*J*_H,H_ = 3.8 Hz, 1H, H-5), 4.34 (dd, ^2^*J*_H,H_ = 11.9 Hz, ^3^*J*_H,H_ = 5.4 Hz, 1H, H-5), 2.36–2.18 (m, 2H, H-3), 2.12 (s, 1H, O-Ac), 1.99 (s, 1H, O-Ac). ^13^C-NMR (125 MHz, D_2_O): *δ* [ppm] = 170.1, 169.5, 166.4 (3C, C=O), 133.4, 129.9, 129.8, 128.5 (4C, C-Aryl), 99.9 (C-1), 79.7 (C-2), 78.8 (C-4), 67.2 (C-5), 32.5 (C-3), 21.7, 21.0 (2C,Me), HR-ESI-MS *m/z* [M + Na]^+^ calcd for C_10_H_13_NaN_3_O_5_^+^: 345.0945, found: 345.0945.



**General Procedure 1: Synthesis of the Nucleosides 2a, 2b and 2c**


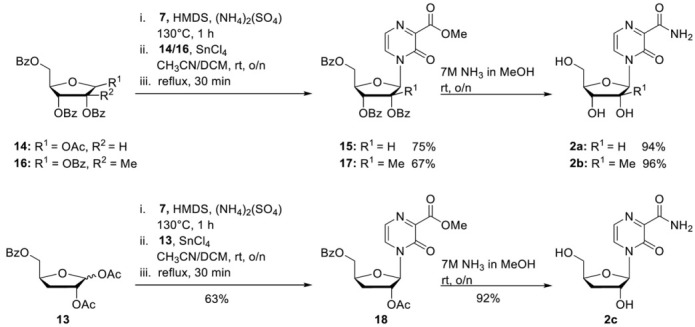



For the synthesis of the nucleosides **2a, 2b,** and **2c,** a slightly modified protocol for the synthesis of **22** was used [29]. **Step 1:** Compound **7** (1.0 eq.) and ammonium sulphate (0.015 eq.) were suspended in hexamethyldisilazane (HMDS) (3 mL/mmol) and heated at 130 °C for 1 h until a clear solution was formed. After all volatiles were removed under reduced pressure, the residue was taken up in dry acetonitrile and CH_2_Cl. The appropriate ribose (0.7–0.8 eq.) followed by tin tetrachloride (1 M solution in CH_2_Cl_2_, 1.0 eq.) was added. The reaction mixture was stirred overnight at room temperature and then refluxed for a further 30 min. After the full conversion of the ribose could be observed, the reaction was quenched by the addition of saturated NaHCO_3_ solution. CH_2_Cl_2_ was added, and the organic layer was separated. The aqueous layer was extracted two times with CH_2_Cl_2_. The combined organic layers were dried over Na_2_SO_4,_ and afterward all volatiles were removed under reduced pressure. Purification of the crude product on silica gel provided the protected nucleosides. **Step 2:** The protected nucleosides from Step 1 were dissolved in ammonia (7 M in MeOH) and stirred overnight at room temperature. After all volatiles were removed under reduced pressure, the residue was dissolved in water. The aqueous layer was washed with CH_2_Cl_2_ three times and once with diethyl ether. After all volatiles were removed under reduced pressure, the crude product was purified on RP18 silica gel using automated flash chromatography (H_2_O/CH_3_CN; 100:0 to 0:100). **T-1106 2a.** According to general procedure 1, with **7** (1.19 g, 7.72 mmol), ammonium sulphate (15.3 mg, 0.12 mmol), and HMDS (23 mL). Then 1-*O*-acetyl-2,3,5-tri-*O*-benzoyl-beta-D-ribofuranose **14** (3.15 g, 6.24 mmol) and tin tetrachloride (7.72 mL, 7.72 mmol) in acetonitrile (55 mL) and CH_2_Cl_2_ (19 mL). Purification of the crude product on silica gel (CH_2_Cl_2_/MeOH; 50:1) provided 2.81 g of **15** (4.68 mmol, 75%) as a colourless foam. ^1^H-NMR (300 MHz, CDCl_3_): δ [ppm] = 8.10–8.06 (m, 2H, H-Bz), 7.98–7.89 (m, 4H, H-Bz), 7.50–7.44 (m, 2H, H-Bz), 7.70 (d, ^3^*J*_H,H_ = 4.4 Hz, 1H, H-5), 7.64–7.44 (m, 5H, H-Bz), 7.41–7.32 (m, 4H, H-6, H-Bz), 6.43 (d, ^3^*J*_H,H_ = 4.7 Hz, 1H, H-1′), 5.91–5.82 (m, 2H, H-2′, H-3′), 4.88 (dd, ^2^*J*_H,H_ = 12.3 Hz, ^3^*J*_H,H_ = 2.8 Hz 1H, H-5′), 4.84–4.81 (m, 1H, H-4′), 4.69 (dd, ^2^*J*_H,H_ = 12.3 Hz, ^3^*J*_H,H_ = 3.8 Hz 1H, H-5′), 3.81 (s, 3H, C*H*_3_-Ester). **15** (716 mg, 1.19 mmol) 7 M ammonia in MeOH (30 mL). Purification on RP18 silica gel using automated flash chromatography (H_2_O/CH_3_CN; 100:0 to 0:100) provided 306 mg of **2a** (1.12 mmol, 94%) as a light-yellow solid. ^1^H-NMR (300 MHz, D_2_O): δ [ppm] = 8.18 (d, ^3^*J*_H,H_ = 4.4 Hz, 1H, H-5), 7.66 (d, ^3^*J*_H,H_ = 4.4 Hz, 1H, H-6), 5.98 (d, ^3^*J*_H,H_ = 1.8 Hz, 1H, H-1′), 4.21 (dd, ^3^*J*_H,H_ = 5.0 Hz, ^3^*J*_H,H_ = 1.8 Hz, 1H, H-2′), 4.18–4.13 (m, 1H, H-4′), 4.07 (dd, ^3^*J*_H,H_ = 8.14 Hz, ^3^*J*_H,H_ = 5.0 Hz, 1H, H-3′), 3.98 (dd, ^2^*J*_H,H_ = 13.1 Hz, ^3^*J*_H,H_ = 2.5 Hz, 1H, H-5′), 3.78 (dd, ^2^*J*_H,H_ = 13.1 Hz, ^3^*J*_H,H_ = 4.2 Hz, 1H, H-5′). All other analytical data are identical to those reported before [27]. **2′-C-Me-T-1106 2b.** According to general procedure 1, with **7** (1.19 g, 7.72 mmol), ammonium sulphate (15.3 mg, 0.12 mmol), and HMDS (23 mL). Then 1,2,3,5-tetra-*O*-benzoyl-2-C-methyl-D-ribofuranose **16** (3.15 g, 5.24 mmol) and tin tetrachloride (7.72 mL, 7.72 mmol) in acetonitrile (55 mL) and CH_2_Cl_2_ (19 mL). Purification of the crude product on silica gel (CH_2_Cl_2_/MeOH; 60:1) provided 1.94 g of **17** (3.63 mmol, 67%) as a colourless foam. ^1^H-NMR (300 MHz, CDCl_3_): δ [ppm] = 8.10–8.06 (m, 4H, H-Bz), 7.85 (d, ^3^*J*_H,H_ = 7.8 Hz, 2H, H-Bz), 7.68 (d, ^3^*J*_H,H_ = 4.3 Hz, 1H, H-5), 7.62–7.54 (m, 2H, H-Bz), 7.51–7.40 (m, 6H, H-Bz, H-6), 7.28–7.23 (m, 2H, H-Bz), 6.85 (s, 1H, H-1′), 5.78 (d, ^3^*J*_H,H_ = 5.2 Hz, 1H, H-4′), 4.92–4.77 (m, 3H, H-5′, H-3′), 3.98 (s, 3H, CH_3_-Ester), 1.66 (s, 3H, CH3-2′). **17** (120 mg, 0.19 mmol) 7 M ammonia in MeOH (5 mL). Purification on RP18 silica gel using automated flash chromatography (H_2_O/CH_3_CN; 100:0 to 0:100) provided 51 mg of **2b** (0.19 mmol, 96%) as a light-yellow solid. ^1^H-NMR (500 MHz, D_2_O): δ [ppm] = 8.25 (d, ^3^*J*_H,H_ = 4.3 Hz, 1H, H-5), 7.78 (d, ^3^*J*_H,H_ = 4.3 Hz, 1H, H-6), 6.33 (s, 1H, H-1′), 4.17–4.15 (m, 1H, H-4′), 4.11–4.08 (m, 1H, H-5′), 3.96–3.90 (m, 2H, H-3′, H-5′), 1.12 (s, 3H, CH_3_-2′). ^13^C-NMR (125 MHz, D_2_O): *δ* [ppm] = 166.2 (C-7), 156.0 (C-3), 143.7 (C-2), 129.4 (C-5), 125.4 (C-6), 92.4 (C-1′), 82.5 (C-4′), 79.7 (C-2′), 72.4 (C-3′), 59.7 (C-5′), 19.3 (*C*H_3_-C-2′). HR-ESI-MS *m/z* [M + H]^+^ calcd for C_11_H_15_NaN_3_O_6_^+^: 308.0859, found: 308.0834.

**3′-Deoxy-T-1106 2c.** According to general procedure 1, with **7** (1.38 g, 8.95 mmol), ammonium sulphate (17.7 mg, 0.13 mmol), and HMDS (25 mL). Then 1,2-*O*-Diacetyl-5-*O*-benzoyl-3-deoxy-D-xylofuranose **13** (2.02 g, 6.26 mmol) and tin tetrachloride (8.95 mL, 8.95 mmol) in acetonitrile (60 mL) and CH_2_Cl_2_ (23 mL). Purification of the crude product on silica gel (CH_2_Cl_2_/MeOH; 60:1) provided 1.64 g of **18** (3.95 mmol, 63) as a colourless foam. ^1^H-NMR (300 MHz, CDCl_3_): δ [ppm] = 8.05–8.02 (m, 2H, *H*-Bz), 7.85 (d, ^3^*J*_H,H_ = 4.3 Hz, 1H, H-5), 7.65–7.59 (m, 1H, *H*-Bz), 7.51–7.45 (m, 2H, *H*-Bz), 7.32 (d, ^3^*J*_H,H_ = 4.3 Hz, 1H, H-6), 6.01 (s, 1H, H-1′), 5.42–5.39 (m, 1H, H-4′), 4.82–4.75 (m, 2H, H-5′, H-2′), 4.62 (dd, ^2^*J*_H,H_ = 13.0 Hz, ^3^*J*_H,H_ = 4.8 Hz, 1H, H-5′), 3.96 (s, 3H, CH_3_-Ester), 2.21–2.17 (m, 2H, H-3′) 2.15 (s, 3H, CH_3_-OAc). **18** (1.63 g, 3.91 mmol) 7 M ammonia in MeOH (30 mL). Purification on RP18 silica gel using automated flash chromatography (H_2_O/CH_3_CN; 100:0 to 0:100) provided 973 mg of **2c** (3.59 mmol, 92%) as a white solid. ^1^H-NMR (500 MHz, D_2_O): δ [ppm] = 8.28 (d, ^3^*J*_H,H_ = 4.3 Hz, 1H, H-5), 7.74 (d, ^3^*J*_H,H_ = 4.3 Hz, 1H, H-6), 5.99 (s, 1H, H-1′), 4.69–4.63 (m, 1H, H-4′), 4.50 (d, ^3^*J*_H,H_ = 4.3 Hz, 1H, H-2′) 4.04 (dd, ^2^*J*_H,H_ = 12.9 Hz, ^3^*J*_H,H_ = 2.8 Hz, 1H, H-5′), 3.83 (dd, ^2^*J*_H,H_ = 12.9 Hz, ^3^*J*_H,H_ = 4.8 Hz, 1H, H-5′), 2.01–1.86 (m, 2H, H-3′). ^13^C-NMR (125 MHz, D_2_O): *δ* [ppm] = 165.9 (C-7), 155.8 (C-3), 142.2 (C-2), 129.0 (C-5), 124.8 (C-6), 93.7 (C-1′), 83.2 (C-4′), 75.8 (C-2′), 61.7 (C-5′), 31.7 (C-3′). HR-ESI-MS *m/z* [M + Na]^+^ calcd for C_10_H_13_NaN_3_O_5_^+^: 278.0753, found: 278.0745.



**Synthesis of the Nucleosides 2d, 2e and 2f**


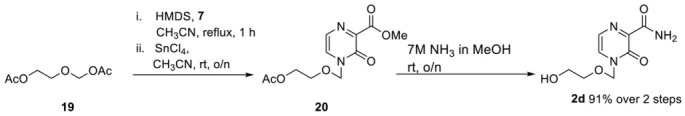



First, **2d** was synthesized with slight modifications as previously published. **7** (555 mg, 3.6 mmol), (2-acetoxyethoxy)methyl acetate (528 mg, 3.0 mmol) and by tin tetrachloride (1 M solution in CH_2_Cl_2_) (3.9 mL 3.6 mmol). Then, 581 mg of **2d** (2.73 mmol, 91%) was received as a yellow oil. The analytical data are identical to those reported before [35].



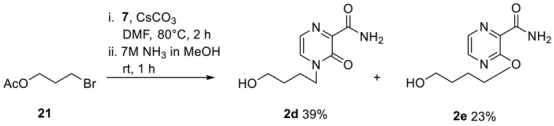



Next, **7** (693 mg, 4.5 mmol) and caesium carbonate (1.47 g, 4.5 mmol) in dry DMF were stirred at room temperature under nitrogen for 30 min. 4-Bromobutyl acetate **21** (433 µL, 3 mmol) was added dropwise via a syringe, and the reaction mixture was stirred at 80 °C for 2 h. Water was added, and the solution was extracted twice with CH_2_Cl_2_. The combined organic layers were washed with water and brine, dried over Na_2_SO_4_, and concentrated under reduced pressure. The crude product was purified by column chromatography (CH_2_Cl_2_/MeOH; 50:1). Pure fraction of the *O*- and *N*-alkylated product were pooled and directly converted into the corresponding carboxamide using step 2 of general procedure 1. 247 mg of **2d** (1.16 mmol, 39%) was received as an orange oil. ^1^H-NMR (300 MHz, DMSO-*d*_6_): δ [ppm] = 8.45 (bs, 1H, CONH_2_), 7.91 (d, ^2^*J*_H,H_ = 4.1 Hz, 1H, H-5), 7.72 (bs, 1H, CONH_2_), 7.50 (d, ^2^*J*_H,H_ = 4.0 Hz, 1H, H-6), 4.46 (bs, 1H, OH), 3.96 (t, ^3^*J*_H,H_ = 7.3 Hz, 2H, H-1′), 3.43–3.38 (m, 2H, H-4′), 1.73–1.68 (m, 2H, H-2′), 1.44–1.38 (m, 2H, H-3′). ^13^C-NMR (125 MHz, DMSO-*d*_6_): δ [ppm] = 164.1 (CONH_2_), 154.7 (C-3), 146.7 (C-2), 133.2 (C-5), 122.9 (C-6), 60.2 (C-4′), 49.2 (C-1′), 29.3 (C-3′), 24.9 (C-2′). ESI-MS *m*/*z* [M + Na]^+^ calcd for C_9_H_13_N_3_O_3_Na^+^: 234.08 [M + Na]^+^, found: 233.9 [M + Na]^+^. 145 mg of **2e** (0.68 mmol, 23%) was received as a colorless solid. ^1^H-NMR (300 MHz, DMSO-*d*_6_): δ [ppm] = 8.29 (d, ^2^*J*_H,H_ = 2.7 Hz, 1H, H-5), 8.19 (d, ^2^*J*_H,H_ = 2.7 Hz, 1H, H-6), 7.88 (bs, 1H, CONH_2_), 7.62 (bs, 1H, CONH_2_), 4.44 (t, ^2^*J*_H,H_ = 5.2 Hz, 1H, OH), 4.34 (t, ^2^*J*_H,H_ = 6.6 Hz, 2H, H-1′), 3.48–3.40 (m, 2H, H-4′), 1.79–1.71 (m, 2H, H-2′), 1.58–1.51 (m, 2H, H-3′). ^13^C-NMR (125 MHz, DMSO-*d*_6_): *δ* [ppm] = 165.7 (CONH_2_), 156.9 (C-3), 142.2 (C-5), 139.0 (C-2), 135.2 (C-6), 66.3 (C-1′), 60.3 (C-4′), 28.9 (C-3′), 25.0 (C-2′). ESI-MS *m/z* [M + Na]^+^ calcd for C_9_H_13_N_3_O_3_Na^+^: 234.08 [M + Na]^+^, found: 233.9 [M + Na]^+^.



**General Procedure 2: Nucleosides 18, 20 and 23**


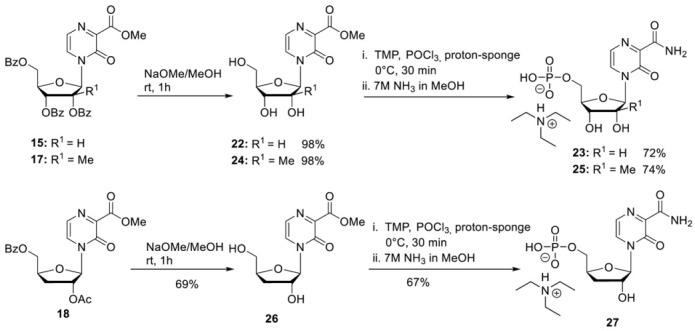



For the synthesis of the nucleosides **2a**, **2b,** and **2c,** a slightly modified protocol for the synthesis of **22** was used [29]. To stir solution of protected nucleoside (1.0 eq.) in dry MeOH, sodium methanolate (5.4 M in MeOH, 0.1 eq.) was added dropwise. After the reaction mixture was stirred for 1 h at room temperature 1 M HCl was added until pH = 6. After all volatiles were removed under reduced pressure, the residue was dissolved in water and washed with CH_2_Cl_2_ four times and with diethyl ether one time. After all volatiles were removed under reduced pressure, the crude product was purified on RP18 silica gel using automated flash chromatography (H_2_O/CH_3_CN; 100:0 to 0:100).

**T-1106 Methyl ester 22.** According to general procedure 2, with **15** (7.30 g, 15.3 mmol) and sodium methanolate (277 µL, 1.50 mmol) in MeOH (70 mL). Purification on RP18 silica gel using automated flash chromatography (H_2_O/CH_3_CN; 100:0 to 0:100) provided 4.32 g of **22** (15.0 mmol, 98%) as a light-yellow solid. ^1^H-NMR (300 MHz, D_2_O): δ [ppm] = 8.34 (d, ^3^*J*_H,H_ = 4.4 Hz, 1H, H-5), 7.71 (d, ^3^*J*_H,H_ = 4.4 Hz, 1H, H-6), 6.06 (d, ^3^*J*_H,H_ = 1.8 Hz, 1H, H-1′), 4.32 (dd, ^3^*J*_H,H_ = 5.0 Hz, ^3^*J*_H,H_ = 1.8 Hz, 1H, H-2′), 4.28–4.23 (m, 1H, H-4′), 4.16 (dd, ^3^*J*_H,H_ = 8.4 Hz, ^3^*J*_H,H_ = 5.0 Hz, 1H, H-3′), 4.07 (dd, ^2^*J*_H,H_ = 13.1 Hz, ^3^*J*_H,H_ = 2.5 Hz, 1H, H-5′), 3.97 (s, 3H, CH_3_-Ester), 3.89 (dd, ^2^*J*_H,H_ = 13.1 Hz, ^3^*J*_H,H_ = 4.2 Hz, 1H, H-5′). All the other data match the literature [29].

**2′-C-Me-T-1106 Methyl ester 24.** According to general procedure 2, with **17** (1.26 g, 2.05 mmol) and sodium methanolate (38 µL, 0.21 mmol) in MeOH (10 mL). Purification on RP18 silica gel using automated flash chromatography (H_2_O/CH_3_CN; 100:0 to 0:100) provided 603 mg of **24** (2.01 mmol, 98%) as a white solid. ^1^H-NMR (300 MHz, D_2_O): δ [ppm] = 8.30 (d, ^3^*J*_H,H_ = 4.3 Hz, 1H, H-5), 7.69 (d, ^3^*J*_H,H_ = 4.3 Hz, 1H, H-6), 6.28 (s, 1H, H-1′), 4.15–4.11 (m, 1H, H-4′), 4.09–4.05 (m, 1H, H-5′), 3.95 (s, 3H, CH_3_-Ester), 3.92–3.87 (m, 2H, H-3′, H-5′), 1.09 (s, 3H, CH_3_-2′). ^13^C-NMR (125 MHz, D_2_O): *δ* [ppm] = 164.8 (C-7), 154.9 (C-3), 142.5 (C-2), 130.5 (C-5), 124.8 (C-6), 92.5 (C-1′), 82.5 (C-4′), 79.3 (C-2′), 72.2 (C-3′), 59.6 (C-5′), 53.6 (-O-*C*H_3_), 19.2 (CH_3_-C-2′).

**3′-Deoxy-T-1106 methyl ester 26.** According to general procedure 2, with **18** (350 mg, 0.84 mmol) and sodium methanolate (15 µL, 0.08 mmol) in MeOH (4 mL). Purification on RP18 silica gel using automated flash chromatography (H_2_O/CH_3_CN; 100:0 to 0:100) provided 175 mg of **26** (0.58 mmol, 69%) as a white solid. ^1^H-NMR (300 MHz, D_2_O): δ [ppm] = 8.36 (d, ^3^*J*_H,H_ = 4.3 Hz, 1H, H-5), 7.71 (d, ^3^*J*_H,H_ = 4.3 Hz, 1H, H-6), 6.01 (s, 1H, H-1′), 4.72–4.67 (m, 1H, H-4′), 4.52 (d, ^3^*J*_H,H_ = 4.3 Hz, 1H, H-2′) 4.08 (dd, ^2^*J*_H,H_ = 12.9 Hz, ^3^*J*_H,H_ = 2.8 Hz, 1H, H-5′), 3.98 (s, 3H, CH_3_-Ester), 3.6 (dd, ^2^*J*_H,H_ = 12.9 Hz, ^3^*J*_H,H_ = 4.8 Hz, 1H, H-5′), 2.03–1.88 (m, 2H, H-3′). ^13^C-NMR (125 MHz, D_2_O): *δ* [ppm] = 164.9 (C-7), 155.2 (C-3), 141.4 (C-2), 130.6 (C-5), 124.7 (C-6), 94.2 (C-1′), 83.6 (C-4′), 76.1 (C-2′), 62.1 (C-5′), 53.6 (-O-*C*H_3_), 32.1 (C-3′).



**General Procedure 3: Synthesis of the 5′-Monophosphates 23, 25 and 27**



**Step 1:** To a stirred solution of nucleoside (1.0 eq.) and proton sponge (2.0 eq.) in trimethyl phosphate (TMP) was added dropwise POCl_3_ at 0 °C. After the full conversion was observed by HPLC, the reaction was quenched by the addition of 1 M TEAB buffer. Afterward the reaction mixture was washed with CH_2_Cl_2_ three times, and all volatiles were removed under reduced pressure. The crude product was purified on RP18 silica gel using automated flash chromatography (H_2_O/CH_3_CN; 100:0 to 0:100). **Step 2:** The intermediate methylester-5′-monophosphate was then dissolved in 7 M NH_3_ in MeOH and stirred over-night at room temperature. After all volatiles were removed under reduced pressure, the crude product was purified on RP18 silica gel using automated flash chromatography (H_2_O/CH_3_CN; 100:0 to 0:100). **T-1106-5′-monophosphate 23. 22** (200 mg, 0.70 mmol) and a proton sponge (299 mg, 1.40 mmol) in TMP (6 mL) were stirred for 30 min. Purification on RP18 silica gel using automated flash chromatography (H_2_O/CH_3_CN; 100:0 to 0:100) provided 290 mg of the methyl-5′-monophosphate carboxylate (0.55 mmol, 78%) as a white hygroscopic solid. The methyl carboxylate intermediate (100 mg, 0.18 mmol) in 7 M NH_3_ in MeOH (5 mL). Purification on RP18 silica gel using automated flash chromatography (H_2_O/CH_3_CN; 100:0 to 0:100) provided 91 mg of **23** (0.17 mmol, 93%) as a white hygroscopic solid. ^1^H-NMR (300 MHz, D_2_O): δ [ppm] = 8.50 (d, ^3^*J*_HH_ = 4.3 Hz, 1H, H-5), 7.83 (d, ^3^*J*_HH_ = 4.3 Hz, 1H, H-6), 6.35 (d, ^3^*J*_HH_ = 1.3 Hz, 1H, H-1′), 4.34–4.31 (m, 3H, H-2′, H-3′, H-4′), 4.27–4.22 (m, 1H, H-5′), 4.08–4.02 (m, 1H, H-5′), 3.21 (q, ^3^*J*_HH_ = 7.3 Hz, 11H, H-a), 1.29 (q, ^3^*J*_HH_ = 7.3 Hz, 17H, H-b). All other analytical data are identical to those reported before [29].

**2′-C-Me-T-1106-5′-monophosphate 25.** According to general procedure 3, **24** (100 mg, 0.33 mmol) and a proton sponge (142 mg, 0.66 mmol) in TMP (3 mL) were stirred for 15 min. Purification on RP18 silica gel using automated flash chromatography (H_2_O/CH_3_CN; 100:0 to 0:100) provided 148 mg of the methyl-5′-monophosphate carboxylate (0.27 mmol, 81%) as a white hygroscopic solid. The methyl carboxylate intermediate (160 mg, 0.29 mmol) in 7 M NH_3_ in MeOH (8 mL). Purification on RP18 silica gel using automated flash chromatography (H_2_O/CH_3_CN; 100:0 to 0:100) provided 152 mg of **25** (0.27 mmol, 92%) as a white hygroscopic solid. ^1^H-NMR (500 MHz, D_2_O): δ [ppm] = 8.43 (d, ^3^*J*_H,H_ = 4.3 Hz, 1H, H-5), 7.83 (d, ^3^*J*_H,H_ = 4.3 Hz, 1H, H-6), 6.35 (s, 1H, H-1′), 4.35–4.31 (m, 1H, H-5′), 4.27–4.24 (m, 1H, H-4′), 4.15–4.09 (m, 2H, H-3′, H-5′), 3.21 (q, ^3^*J*_H,H_ = 7.3 Hz, 10H, H-a), 1.29 (q, ^3^*J*_H,H_ = 7.3 Hz, 15H, H-b), 1.14 (s, 3H, C*H*_3_-2′). ^31^P-NMR (162 MHz, D_2_O): δ [ppm] = 2.05. ^13^C-NMR (125 MHz, D_2_O): *δ* [ppm] = 166.4 (C-7), 156.1 (C-3), 143.4 (C-2), 129.9 (C-5), 125.8 (C-6), 92.3 (C-1′), 81.6 (C-4′), 79.9 (C-2′), 71.6 (C-3′), 62.2 (C-5′), 19.2 (CH_2_-2′). HR-ESI-MS *m/z* calcd for C_11_H_15_N_3_O_9_P^−^: 364.0551 [M − H]^−^, found.: 364.0557 [M − H]^−^.

**3′-Deoxy-T-1106-5′monophosphate 27.** According to general procedure 3, **26** (152 mg, 0.56 mmol), proton sponge (241 mg, 0.66 mmol), and POCl_3_ (205 µL, 1.12 mmol) in TMP (5 mL) were stirred for 15 min. Purification on RP18 silica gel using automated flash chromatography (H_2_O/CH_3_CN; 100:0 to 0:100) provided 240 mg of the methyl-5′-monophosphate carboxylate (0.42 mmol, 75%) as a white hygroscopic solid. The methyl carboxylate intermediate (240 mg, 0.42 mmol) in 7 M NH_3_ in MeOH (12 mL). Purification on RP18 silica gel using automated flash chromatography (H_2_O/CH_3_CN; 100:0 to 0:100) provided 205 mg of **27** (0.38 mmol, 90%) as a white hygroscopic solid. ^1^H-NMR (500 MHz, D_2_O): δ [ppm] = 8.43 (d, ^3^*J*_H,H_ = 4.3 Hz, 1H, H-5), 7.82 (d, ^3^*J*_H,H_ = 4.3 Hz, 1H, H-6), 6.04 (s, 1H, H-1′), 4.83–4.80 (m, 1H, H-4′), 4.55 (d, ^3^*J*_H,H_ = 4.3 Hz, 1H, H-2′) 4.42–4.38 (m, 1H, H-5′), 4.13–4.09 (m, 1H, H-5′), 3.22 (q, ^3^*J*_H,H_ = 7.3 Hz, 8H, H-a), 2.14–2.02 (m, 2H, H-3′), 1.28 (t, ^3^*J*_H,H_ = 7.3 Hz, 12H, H-b). ^31^P-NMR (162 MHz, D_2_O): δ [ppm] = 0.21. ^13^C-NMR (125 MHz, D_2_O): *δ* [ppm] = 164.3 (C-7), 156.3 (C-3), 142.4 (C-2), 129.8 (C-5), 125.5 (C-6), 93.6 (C-1′), 82.1 (C-4′), 76.2 (C-2′), 64.9 (C-5′), 31.8 (C-3′). HR-ESI-MS *m/z* [M + Na]^+^ calcd for C_11_H_15_N_3_O_9_P^−^: 364.0551 [M − H]^−^, found: 364.0557 [M − H].


**General Procedure 4: Synthesis of T-1106-Tri*PPP*ros 4a–4f**


**Step 1:** To a stirred solution of *H*-phosphonate (1.0 eq.) in dry CH_3_CN (10 mL) *N*-chlorosuccinimide (2.5 eq.) was added. After the reaction mixture was stirred overnight at 50 °C, tetrabutylammonium phosphate monobasic solution (0.4 M in acetonitrile) (3.0 equiv.) was added dropwise. After the reaction mixture was stirred for one more hour, all volatiles were removed under reduced pressure, and the residue was dissolved in CH_2_Cl_2_ and subsequently washed with 1 M ammonium acetate solution and cold water. Afterward, all volatiles were removed under reduced pressure to afford the corresponding pyrophosphate in quantitative yield. **Step 2:** The corresponding pyrophosphate was dissolved in dry CH_3_CN (4 mL) and cooled to 0 °C. Afterward, a mixture of trifluoroacetic anhydride (5.0 equiv.) and Et_3_N (8.0 equiv.) in dry CH_3_CN (1.5 mL) was added dropwise. After the reaction mixture was stirred for 10 min at room temperature, all volatiles were removed under reduced pressure. Afterward, the residue was dissolved in dry CH_3_CN (6 mL) and cooled to 0 °C, and 1-methylimidazole was added. The reaction mixture was stirred for 10 min before the NMP (0.3 eq.) in dry DMF (2.5 mL) was added. The reaction mixture was stirred for 2 h at room temperature before all volatiles were removed under reduced pressure. The desired target molecules could finally be obtained after purification with RP18 silica gel using automated flash chromatography (H_2_O/CH_3_CN; 100:0 to 0:100) followed by ion-exchange to the ammonium form with Dowex 50WX8 cation-exchange resin and a second chromatography step with RP18 silica gel using automated flash chromatography (H_2_O/CH_3_CN; 100:0 to 0:100).



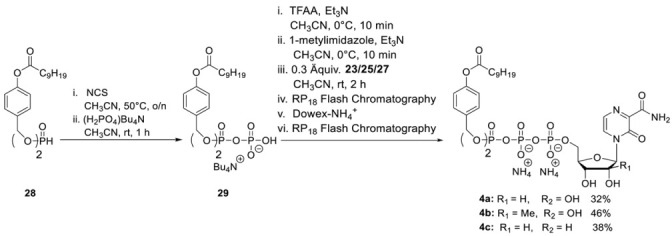



**T-1106-C9C9-Tri*PPP*ro 4a.** According to general procedure 4, **28** (290 mg, 0.49 mmol) and **21** (50 mg, 0.12 mmol) provided 40 mg of **4a** (0.04 mmol, 32%) as a white cotton. The analytical data are identical to those reported [29].

**2′-C-Me-T-1106- C9C9-Tri*PPP*ro 4b.** According to general procedure 4, with **28** (290 mg, 0.49 mmol) and **25** (70 mg, 0.12 mmol) provided 60 mg of **4b** (0.05 mmol, 46%) as a white cotton. ^1^H-NMR (500 MHz, MeOD): δ [ppm] = 8.43 (d, ^3^*J*_H,H_ = 4.3 Hz, 1H, H-5), 7.83 (d, ^3^*J*_H,H_ = 4.3 Hz, 1H, H-6), 7.40–7.39 (m, 4H, H-3″), 7.04–7.02 (m, 4H, H-2″), 6.27 (s, 1H, H-1′), 5.20–5.14 (m, 4H, Ph-CH_2_), 4.49–4.47 (m, 1H, H-5′), 4.42–4.39 (m, 1H, H-4′), 4.15–4.13 (m, 1H, H-5′), 4.08 (d, ^3^*J*_H,H_ = 9.3 Hz, 1H, H-3′), 2.58 (t, ^3^*J*_H,H_ = 7.4 Hz, 4H, H-b′), 1.74 (p, ^3^*J*_H,H_ = 7.3 Hz, 4H, H-c′), 1.46–1.32 (m, 24H, H-d′, H-e′, H-f′, H-g′, H-h′, H-i′), 1.06 (s, 3H, C*H*_3_-2′), 0.92 (t, ^3^*J*_H,H_ = 6.8 Hz, 6H, H-j′). ^13^C-NMR (125 MHz, MeOD): *δ* [ppm] = 173.4 (C-a′), 166.3 (C-7), 156.0 (C-3), 150.1 (C-4″), 143.4 (C-2), 134.9 (d, ^3^*J*_CP_ = 6.4 Hz, C-1″), 130.7 (C-5), 130.5 (C-3″), 126.2 (C-6), 122.8 (C-2″), 93.4 (C-1′), 83.1 (d, ^3^*J*_CP_ = 9.2 Hz, C-4′), 80.5 (C-2′), 72.3 (C-3′), 70.4 (d, ^2^*J*_C,P_ = 5.8 Hz, Ph-*C*H_2_), 64.3 (d, ^2^*J*_C,P_ = 5.3 Hz, C-5′), 35.0 (C-b′), 33.0 (C-c′), 30.6 (C-d′), 30.5 (C-e′), 30.4 (C-f′), 30.2 (C-g′), 26.0 (C-h′), 23.8 (C-i′), 20.1 (-O-CH_3_), 14.5 (C-j′). ^31^P-NMR (162 MHz, MeOD): δ [ppm] = -11.65 (d, ^2^*J*_P,P_ = 19.4 Hz, P-α), −13.23 (d, ^2^*J*_P,P_ = 17.0 Hz, P-γ), −23.68 (d, ^2^*J*_P,P_ = 18.6 Hz, P-β). HR-ESI-MS *m/z* [M + H]^+^ calcd for C_45_H_67_N_3_O_19_P_3_^−^: 1046.3582 [M + H]^+^, found: 1046.3644 [M + H]^+^.

**3′-Deoxy-T-1106- C9C9-Tri*PPP*ro 4c.** According to general procedure 4, with **28** (310 mg, 0.52 mmol) and **27** (55 mg, 0.13 mmol) provided 51 mg of **4c** (0.05 mmol, 38%) as a white cotton. ^1^H-NMR (500 MHz, MeOD): δ [ppm] = 8.47 (d, ^3^*J*_H,H_ = 4.3 Hz, 1H, H-5), 7.82 (d, ^3^*J*_H,H_ = 4.3 Hz, 1H, H-6), 7.40–7.39 (m, 4H, H-3″), 7.04–7.02 (m, 4H, H-2″), 6.27 (s, 1H, H-1′), 5.18–5.16 (m, 4H, Ph-CH_2_), 4.70–4.67 (m, 1H, H-4′), 4.53–4.50 (m, 1H, H-5′), 4.36 (d, ^3^*J*_H,H_ = 4.6 Hz, 1H, H-2′), 4.27–4.23 (m, 1H, H-5′), 2.57 (t, ^3^*J*_H,H_ = 7.4 Hz, 4H, H-b′), 2.10–2.05 (m, 1H, H-3′), 1.91–1.88 (m, 1H, H-3′), 1.73 (p, ^3^*J*_H,H_ = 7.3 Hz, 4H, H-c′), 1.46–1.27 (m, 24H, H-d′, H-e′, H-f′, H-g′, H-h′, H-i′), 0.93 (t, ^3^*J*_H,H_ = 6.8 Hz, 6H, H-j′). ^13^C-NMR (125 MHz, MeOD): *δ* [ppm] = 173.7 (C-a′), 166.4 (C-7), 157.1 (C-3), 152.3 (C-4″), 143.3 (C-2), 134.9 (d, ^3^*J*_C,P_ = 6.4 Hz, C-1″), 130.7 (C-5), 130.5 (C-3″), 126.5 (C-6), 122.8 (C-2″), 95.3 (C-1′), 82.9 (d, ^3^*J*_C,P_ = 9.2 Hz, C-4′), 77.5 (C-2′), 70.3 (d, ^2^*J*_C,P_ = 5.3 Hz, ^4^*J*_C,P_ = 2.8 Hz, C-5′), 66.6 (d, ^2^*J*_C,P_ = 5.8 Hz, Ph-CH_2_), 64.3 (d, ^2^*J*_C,P_ = 5.3 Hz, C-5′), 35.0 (C-b′), 33.3 (C-3′), 33.0 (C-c′), 30.6 (C-d′), 30.5 (C-e′), 30.4 (C-f′), 30.2 (C-g′), 26.0 (C-h′), 23.8 (C-i′), 14.5 (C-j′). ^31^P-NMR (162 MHz, MeOD): δ [ppm] = −11.62 (d, ^2^*J*_P,P_ = 19.6 Hz, P-α), −13.25 (d, ^2^*J*_P,P_ = 17.3 Hz, P-γ), −23.80 (d, ^2^*J*_P,P_ = 18.9 Hz, P-β). HR-ESI-MS *m/z* [M + H]^+^ calcd for C_44_H_63_N_3_O_18_P_3_^−^: 1016.3476 [M + H]^+^, found: 1016.3544 [M + H]^+^.



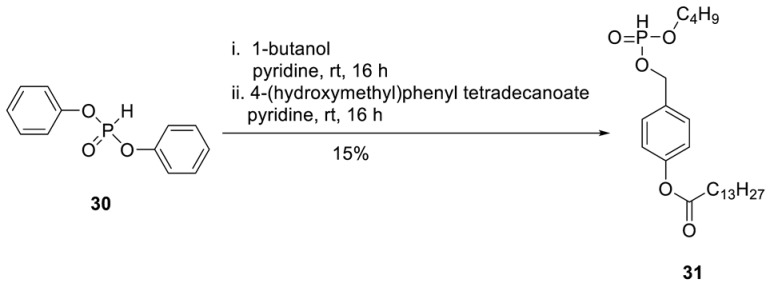



**H-Phosphonate 31.** To a stirred solution of diphenylphosphite **30** (2.37 g, 10.1 mmol) in dry pyridine (20 mL), 1-butanol (610 µL, 6.75 mmol) was added at 0 °C. After the reaction mixture was stirred overnight at room temperature, a solution of 4-(hydroxymethyl)phenyltetradecanoate (2.71 g, 8.10 mmol) in pyridine (10 mL) was added dropwise. After the reaction mixture was stirred overnight at room temperature, all volatiles were removed under reduced pressure, and the crude product was purified on silica gel (CH_2_Cl_2_/MeOH/; 30:1 1% AcOH). A second chromatography step on silica gel using automated flash chromatography (CH_2_Cl_2_/3% acetone) provided 0.45 g of **31** (0.99 mmol, 15%) as a colourless wax. ^1^H-NMR (500 MHz, MeOD): δ [ppm] = 7.41 (dt, ^3^*J*_H,H_ = 8.6 Hz, ^4^*J*_H,H_ = 2.1 Hz, 2H, H-2), 7.10 (dt, ^3^*J*_H,H_ = 8.5 Hz, ^4^*J*_H,H_ = 2.0 Hz, 2H, H-3), 6.87 (d, ^3^*J*_H,P_ = 700 Hz, 1H, P-*H*) 5.10 (d, ^3^*J*_H,H_ = 9.6 Hz, 2H, PH-CH_2_), 4.12–3.96 (m, 2H, H-a′), 2.55 (t, ^3^*J*_H,H_ = 7.5 Hz, H-b), 1.80–1.70 (m, 2H, H-c), 1.70–1.60 (m, 2H, H-b′), 1.44–1.26 (m, 22H, H-d, H-e, H-f, H-g, H-h, H-i, H-j, H-k, H-l, H-m, H-c′), 0.93–0.87 (m, 6H, H-n, H-d′). ^13^C-NMR (125 MHz, MeOD): *δ* [ppm] = 172.3 (C-a), 151.1 (C-1), 133.3 (d, ^3^*J*_C,P_ = 6.1 Hz, C-4), 129.3 (C-2), 122.0 (C-3), 66.7 (d, ^2^*J*_C,P_ = 5.5 Hz, Ph-CH_2_), 65.9 (d, ^2^*J*_C,P_ = 6.3 Hz, C-a′), 34.5 (C-b), 32.5 (d, ^3^*J*_C,P_ = 6.2 Hz, C-b′), 32.0, 29.9, 29.8, 29.7, 29.6, 29.5, 29.4, 29.3, (C-d, C-e, C-f, C-g, C-h, C-i, C-j, C-k, C-l), 25.1 (C-c), 22.8 (C-c′), 18.8 (C-m), 14.3 (C-n), 13.7 (C-d′). ^31^P-NMR (162 MHz, MeOD): δ [ppm] = 8.93. HR-ESI-MS *m/z* [M + H]^+^ calcd for C_25_H_44_O_5_P: 454.2848 [M + H]^+^, found: 455.2625 [M + H]^+^.



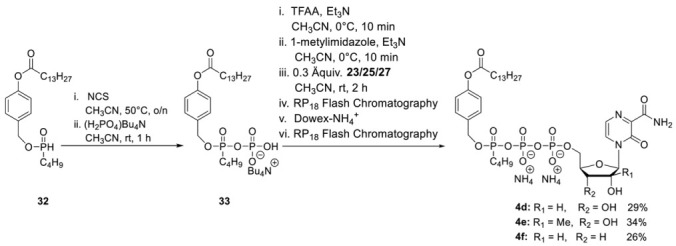



**T-1106-C4alkylC9-Tri*PPP*ro 4d.** According to general procedure 4, **33** (250 mg, 0.55 mmol) and **19** (50 mg, 0.12 mmol) provided 32 mg of **4d** (0.04 mmol, 29%) as a white cotton. ^1^H-NMR (500 MHz, MeOD): δ [ppm] = 8.44 (dd, ^3^*J*_H,H_ = 1.0Hz, ^3^*J*_H,H_ = 4.3 Hz, 1H, H-5), 7.84 (dd, ^3^*J*_H,H_ = 2.4 Hz, ^3^*J*_H,H_ = 4.3 Hz, 1H, H-6), 7.49–7.47 (m, 2H, H-3″), 7.08–7.07 (m, 2H, H-2″), 6.07 (s, 1H, H-1′), 5.24–5.20 (m, 2H, Ph-CH_2_), 4.46–4.44 (m, 1H, H-5′), 4.37–4.32 (m, 2H, H-2′, H-4′), 4.25–4.24 (m, 1H, H-5′), 4.21–4.19 (m, 1H, H-3′), 4.16–4.11 (m, 2H, H-a″) 2.60–2.57 (m, 2H, H-b′), 1.76–1.71 (m, 2H, H-c′), 1.65–1.59 (m, 2H, H-b″), 1.45–1.30 (m, 22H, H-d′, H-e′, H-f′, H-g′, H-h′, H-i′, H-j′, H-k′, H-l′, H-m′, H-c″), 0.93–0.89 (m, 6H, H-n, H-d′). ^13^C-NMR (125 MHz, MeOD): *δ* [ppm] = 173.7 (C-a′), 166.3 (C-7), 157.1 (C-3), 152.2 (C-4″), 143.7 (C-2), 135.1 (d, ^3^*J*_C,P_ = 7.4 Hz, C-1″), 130.8 (C-5), 130.3 (C-3″), 126.1 (C-6), 122.8 (C-2″), 92.3 (C-1′), 84.6 (d, ^3^*J*_C,P_ = 9.2 Hz, C-4′), 76.9 (C-2′), 70.2 (C-3′), 69.5 (d, ^2^*J*_C,P_ = 6.6 Hz, Ph-*C*H_2_), 69.4 (d, ^2^*J*_C,P_ = 3.9 Hz, C-a″), 64.9 (m, C-5′), 35.0 (C-b′), 33.3 (d, ^3^*J*_C,P_ = 7.4 Hz, C-b″), 33.1, 30.9, 30.8, 30.7, 30.6, 30.5, 30.4, 30.2, (C-d′, C-e′, C-f′, C-g′, C-h′, C-i′, C-j′, C-k′, C-l′), 25.9 (C-c′), 23.7 (C-c″), 19.7 (C-m′), 14.4 (C-n′), 13.9 (C-d″). ^31^P-NMR (162 MHz, MeOD): δ [ppm] = −11.73 (dd, ^2^*J*_P,P_ = 19.4 Hz, ^3^*J*_P,P_ = 2.5 Hz, P-α), −13.07 (dd, ^2^*J*_P,P_ = 17.0 Hz, ^3^*J*_P,P_ = 2.5 Hz, P-γ), −23.75 (d, ^2^*J*_P,P_ = 18.2 Hz, P-β). HR-ESI-MS *m/z* [M + H]^+^ calcd for C_35_H_54_N_3_O_17_P_3_^+^ 884.2901 [M + H]^+^, found: 884.2946 [M + H]^+^.

**2′-C-Me-T-1106-C4alkylC9-Tri*PPP*ro 4e.** According to general procedure 4, with **33** (250 mg, 0.55 mmol) and **19** (74 mg, 0.13 mmol) provided 38 mg of **4e** (0.04 mmol, 34%) as a white cotton. ^1^H-NMR (500 MHz, MeOD): δ [ppm] = 8.44 (dd, ^3^*J*_H,H_ = 1.2 Hz, ^3^*J*_H,H_ = 4.3 Hz, 1H, H-5), 7.84 (dd ^3^*J*_H,H_ = 2.4 Hz, ^3^*J*_H,H_ = 4.3 Hz, 1H, H-6), 7.49–7.47 (m, 2H, H-3″), 7.08–7.07 (m, 2H, H-2″), 6.29 (s, 1H, H-1′), 5.24–5.18 (m, 2H, Ph-CH_2_), 4.48–4.46 (m, 1H, H-5′), 4.41–4.38 (m, 1H, H-4′), 4.16–4.11 (m, 3H, H-5′, H-a″), 4.09–4.08 (dd, ^3^*J*_H,H_ = 2.0 Hz, ^3^*J*_H,H_ = 9.3 Hz, 1H, H-3′), 2.58 (t, ^3^*J*_H,H_ = 7.4 Hz, 2H, H-b′), 1.74 (p, ^3^*J*_H,H_ = 7.3 Hz, 2H, H-c′), 1.64–1.59 (m, 2H, H-b″), 1.45–1.30 (m, 22H, H-d′, H-e′, H-f′, H-g′, H-h′, H-i′, H-j′, H-k′, H-l′, H-m′, H-c″), 1.08 (d, ^3^*J*_H,H_ = 6.4 Hz, 3H, C*H*_3_-2′), 0.92–0.89 (m, 6H, H-n, H-d′). ^13^C-NMR (125 MHz, MeOD): *δ* [ppm] = 173.7 (C-a′), 166.2 (C-7), 157.0 (C-3), 153.2 (C-4″), 144.2 (C-2), 135.1 (dd, ^3^*J*_C,P_ = 7.4 Hz, ^3^*J*_CP_ = 2.4 Hz, C-1″), 130.7 (C-5), 130.4 (C-3″), 126.2 (C-6), 122.8 (C-2″), 93.4 (C-1′), 83.1 (d, ^3^*J*_C,P_ = 9.2 Hz, C-4′), 80.5 (C-2′), 72.3 (C-3′), 70.2 (d, ^2^*J*_C,P_ = 5.8 Hz, Ph-*C*H_2_), 69.5 (d, ^2^*J*_C,P_ = 6.4 Hz, C-a″), 64.3 (d, ^2^*J*_C,P_ = 5.3 Hz, C-5′), 35.0 (C-b′), 33.2 (d, ^3^*J*_C,P_ = 7.4 Hz, C-b″), 33.1, 30.9, 30.8, 30.7, 30.6, 30.5, 30.4, 30.2, (C-d′, C-e′, C-f′, C-g′, C-h′, C-i′, C-j′, C-k′, C-l′), 25.9 (C-c′), 23.7 (C-c″), 20.2 (-O-*C*H_3_), 19.7 (C-m′), 14.4 (C-n′), 13.9 (C-d″). ^31^P-NMR (162 MHz, MeOD): δ [ppm] = −11.65 (dd, ^2^*J*_P,P_ = 19.8 Hz, ^3^*J*_P,P_ = 3.4 Hz, P-α), −13.07 (dd, ^2^*J*_P,P_ = 17.2 Hz, ^3^*J*_P,P_ = 5.5 Hz, P-γ), −23.74 (d, ^2^*J*_P,P_ = 18.2 Hz, P-β). HR-ESI-MS *m/z* [M + H]^+^ calcd for C_36_H_56_N_3_O_17_P_3_^−^: 898.3057 [M + H]^+^, found: 898.3110 [M + H]^+^.

**3′-Deoxy-T-1106-C4alkylC9-Tri*PPP*ro 4f** According to general procedure 4, with **33** (250 mg, 0.55 mmol) and **19** (60 mg, 0.14 mmol) provided 30 mg of **4f** (0.04 mmol, 26%) as a white cotton. ^1^H-NMR (500 MHz, MeOD): δ [ppm] = 8.50 (d, ^3^*J*_H,H_ = 4.3 Hz, 1H, H-5), 7.83 (d, ^3^*J*_H,H_ = 2.0 Hz, ^3^*J*_H,H_ = 4.3 Hz, 1H, H-6), 7.49–7.47 (m, 2H, H-3″), 7.08–7.07 (m, 2H, H-2″), 5.93 (s, 1H, H-1′), 5.24–5.18 (m, 2H, Ph-CH_2_), 4.71–4.46 (m, 1H, H-4′), 4.53–4.50 (m, 1H, H-5′), 4.36 (d, ^3^*J*_H,H_ = 4.6 Hz, 1H, H-2′), 4.27–4.23 (m, 1H, H-5′), 4.16–4.12 (m, 2H, H-a″), 2.57 (t, ^3^*J*_H,H_ = 7.5 Hz, 2H, H-b′), 2.10–2.05 (m, 1H, H-3′), 1.92 (dd, ^3^*J*_H,H_ = 14.0 Hz, ^2^*J*_H,H_ = 5.2 Hz, 1H, H-3′), 1.72 (p, ^3^*J*_H,H_ = 7.3 Hz, 2H, H-c′), 1.65–1.60 (p, ^3^*J*_H,H_ = 7.3 Hz, 2H, H-b″), 1.45–1.30 (m, 22H, H-d′, H-e′, H-f′, H-g′, H-h′, H-i′, H-j′, H-k′, H-l′, H-m′, H-c″), 0.92–0.89 (m, 6H, H-n, H-d′). ^13^C-NMR (125 MHz, MeOD): *δ* [ppm] = 173.7 (C-a′), 166.4 (C-7), 157.2 (C-3), 152.3 (C-4″), 143.3 (C-2), 135.1 (d, ^3^*J*_C,P_ = 7.4 Hz, ^4^*J*_C,P_ = 2.4 Hz, C-1″), 130.8 (C-5), 130.3 (C-3″), 126.0 (C-6), 122.8 (C-2″), 95.3 (C-1′), 82.9 (d, ^3^*J*_C,P_ = 9.2 Hz, C-4′), 77.5 (C-2′), 70.2 (m, C-5′), 66.5 (dd, ^2^*J*_C,P_ = 5.8 Hz, ^3^*J*_C,P_ = 1.9 Hz, Ph-*C*H_2_), 66.5 (d, ^2^*J*_C,P_ = 5.7 Hz, C-5′), 35.0 (C-b′), 33.3 (d, ^3^*J*_C,P_ = 7.4 Hz, C-b″), 33.1, 30.9, 30.8, 30.7, 30.6, 30.5, 30.4, 30.2, (C-d′, C-e′, C-f′, C-g′, C-h′, C-i′, C-j′, C-k′, C-l′), 26.0 (C-c′), 23.7 (C-c″), 19.7 (C-m′), 14.4 (C-n′), 14.0 (C-d″). ^31^P-NMR (162 MHz, MeOD): δ [ppm] = −11.65 (d, ^2^*J*_P,P_ = 19.4 Hz, P-α), −13.07 (dd, ^2^*J*_P,P_ = 17.2 Hz, ^3^*J*_P,P_ = 6.5 Hz, P-γ), −23.75 (d, ^2^*J*_P,P_ = 18.2 Hz, P-β).

Detailed structural information on the compounds, including characterization by NMR spectroscopy and HPLC analysis, is provided elsewhere [29]. T-705 and all its derivatives were dissolved in DMSO (Sigma-Aldrich, St. Louis, MO, USA), and stock solutions were aliquoted and stored at −20 °C. Oseltamivir (Sigma-Aldrich, St. Louis, MO, USA) was dissolved in double-distilled water, aliquoted, and stored at −20 °C.

### 2.6. Virus Titration by Plaque Assay

Titration of infectious virus particles was carried out on MDCKII cells in 12-well tissue culture plates. Infectious cell culture supernatants were serially diluted 10-fold in PBS. Cells were washed once with PBS, and 150 µL of the appropriated dilution was added to the respective well. Cells were incubated for 30 min at 37 °C and subsequently covered with a 1:1 mixture of 2.5% Avicel (in double-distilled water) and 2× MEM medium (Fisher Scientific, Hampton, NH, USA), supplemented with 2% Penicillin/Streptomycin, 2% l-glutamine, 0.4% BSA and 1 µg/mL TPCK-treated trypsin. After incubation for 72 h at 37 °C, cells were washed once with PBS and then fixed with 4% paraformaldehyde solution for 30 min at 4 °C. Staining of H1N1 IAV plaques was performed with a primary influenza A nucleoprotein specific antibody (Abcam, Cambridge, UK) and a secondary HRP-conjugated antibody (Southern Biotech, Birmingham, AL, USA). Plaques were visualized by adding an HRP substrate (Seracare, Milford, MA, USA). H3N2, H5N1, and H7N9 IAV plaques were stained by counterstaining with crystal violet solution. Infectious virus titers were determined as plaque-forming units per ml (pfu/mL).

### 2.7. Cell Viability Assay

Measurement of cell viability upon compound treatment was performed using the CellTiter 96^®^ Non-Radioactive Cell Proliferation Assay (MTT) (Promega, Madison, WI, USA) according to the manufacturer’s instructions. In short, 1 × 10^4^ MDCKII cells per well were seeded in 96-well plates. The cells were treated for 24 h with eight different inhibitor concentrations, diluted in an MDCKII cell culture medium. As a vehicle control, cells were treated with DMSO only. After incubation of the cells with the MTT-like substrate for 4 h, a stop solution was added, and the plates were further incubated at 37 °C overnight to allow solubilization of the blue crystals. Absorbance at 570 nm was measured on a Saphire2 plate reader (TECAN, Männedorf, Switzerland), and the cell viability was calculated in comparison to the DMSO control treatment.

### 2.8. Quantification and Statistical Analyses

All data generated in this study were analyzed and graphically visualized using Microsoft Excel (2016) and GraphPad PRISM Version 9.4.0 (RRID: SCR_002798). Statistically significant differences between vehicle controls and single-drug treatments at a defined concentration were determined using the Kruskal-Wallis test in combination with Dunn’s multiple comparisons tests.

In order to investigate synergistic interactions of compound **3a** and oseltamivir, the open-source web application SynergyFinder Plus was used [36]. Herein, four different reference models were employed to evaluate drug synergy: (i) the Bliss independence model (which expects that two drugs induce independent effects), (ii) the Zero Interaction Potency (ZIP) model (compares changes in the potency of the dose-response curves between individual drugs and their combinations), (iii) the highest single-agent (HSA) model (assumes that the expected effect of the combination treatment equals the maximum of the single drug responses), and (iv) the Loewe additivity (Loewe) model (which calculates the expected response in case both drugs were identical).

## 3. Results

### 3.1. Chemical Synthesis of a T-1105/1106 Derivate Library Consisting of Nucleosides, Di- and Triphosphate Pronucleotides

In this study, we synthesized a library of T-705 derivates in order to identify new variants with increased antiviral activity against both seasonal and especially highly pathogenic avian influenza viruses (Figure 1). Herein, we pursued two different approaches: First, we developed DP (diphosphate) and TP (triphosphate) prodrugs of the T-1105 base, aiming to circumvent any cellular bottlenecks in nucleobase analogue activation (e.g., HGPRT and nucleotide kinase activity) (Figure 1). In order to achieve cell permeability of the negatively charged nucleotides, we introduced lipophilic masks at the terminal phosphate. After uptake into the cells, the release of the nucleotides is initiated by nucleophiles or esterase/lipase activity [29]. The released nucleoside analogue TPs can directly serve as a substrate for the viral RNA-dependent RNA polymerase (RdRP), whereas the DPs either act directly as substrates of the polymerase [29] or need to undergo further phosphorylation via cellular kinases (Figure 1).

Since the main mode of action of T-1106 is lethal mutagenesis, we aimed to modify the ribose part of T-1106 to enable chain termination as the predominant mechanism of action that might lead to enhanced antiviral activity. For that reason, we chose two different modifications, the 2′-C-methyl- and the 3′-deoxymodification. After a nucleoside analogue with a 2′-C-methylmodification is incorporated into the viral RNA strand, it prevents the RdRP from fluctuating into its closed conformation, which is necessary for the incorporation of the next incoming NTP (nucleotide triphosphate) [37]. The removal of the 3′-OH-group should lead to obligate chain termination due to the fact that the next incoming NTP has no functional group that it can bind. In addition to the modifications of the ribose part, we introduced a non-cleavable alkyl modification at the γ-phopshate group of the TP prodrugs. A previous study has shown that γ-alkyl-NTPs have a higher selectivity towards viral polymerases, which might reduce undesired cytotoxicity and side effects of the nucleoside analogues [38].

### 3.2. T-1106 Di- and Triphosphate Prodrugs Potently Inhibit Seasonal Influenza Virus Replication

The synthesis and antiviral activity of T-1105-derived DP and TP prodrugs against influenza A (A/X-31 (H3N2)) and influenza B (B/Ned/537/05) viruses were originally described by Huchting et al. in 2018 [29]. Here, anti-influenza activity was measured by microscopic scoring of virus-induced cytopathic effects. In order to assess the potency of our T-705 derivate library to block the production of infectious virus progeny, we infected MDCKII cells with a seasonal H3N2 IAV isolate and determined virus titers in the presence of drug treatment. Reference compounds **1a** (T-705) and **1b** (T-1105) reduced virus replication by approximately 1.5 logs, in good agreement with previous reports (Figure 2a) [28,29]. T-1106 **2a** showed slightly higher activity compared to T-1105 **1b**. Surprisingly, any chemical modifications in the T-1106 nucleosides **2b**–**2f** completely abrogated antiviral activity (Figure 2a). In contrast, at 50 µM, the DP prodrugs **3a** and **3b** reduced viral titers to the limit of detection, indicating high therapeutic efficacy. We further observed a pronounced difference in the antiviral activity of the TP prodrugs: while treatment with compound **4a** showed a modest reduction in viral replication, albeit without reaching statistical significance, for TP compounds **4b**–**4f**, no antiviral activity was detected (Figure 2a). Overall, these data confirm that T-1106-DP prodrugs **3a** and **3b** are highly active against H3N2 IAV replication in vitro, likely by overcoming bottlenecks in the intracellular conversion of T-705 **1a** and T-1105 **1b** to the metabolically active forms.

In order to further characterize the antiviral properties of our lead DP (**3a**, **3b)** and TP (**4a**) prodrug candidates, we determined their inhibitory concentration 50 (IC_50_) against a more recent H1N1 IAV clinical isolate (Figure 2b–e). Compared to reference compound **1a** (T-705), DP prodrug **3a** showed a 10-fold increase in antiviral activity (IC_50_ of 1.29 vs. 0.13 µM, respectively) (Figure 2b,c; Table 1). DP and TP prodrug compounds **3b** and **4a** inhibited H1N1 IAV replication with similar IC_50_ values as reference compound **1a** (IC_50_ of 1.49 and 3.32 µM, respectively; Figure 2d,e; Table 1). Importantly, inhibition of viral replication at these concentrations was not caused by impaired cell viability upon drug treatment (Figure 2b–e; Table 1). It should be noted that as for H3N2 IAV (Figure 2a), maximum inhibition of H1N1 replication to the levels of detection limit was only achieved with the DP and TP prodrugs, but not with reference compound T-705 **1a** (Appendix A,g,j).

In summary, both DP and TP prodrugs are potent inhibitors of seasonal H1N1 and H3N2 IAV replication, with comparable or increased antiviral activity as the original and clinically licensed T-705 **1a**.

### 3.3. T-1106 Prodrugs ***3a***, ***3b*** and ***4a*** Are Highly Active against H5N1 and H7N9 HPAIV

Next, we sought to analyze whether our lead DP and TP prodrug candidates (**3a**/**3b** and **4a**, respectively) could also inhibit the replication of HPAIV (H5N1, H7N9) that have crossed the species barrier in the past and caused mortality rates of up to 40% in humans. Most importantly, these viruses are currently closely monitored by the WHO for the emergence of novel strains with pandemic potential. T-705 **1a** inhibited H5N1 replication with an IC_50_ of 1.14 µM, while DP and TP prodrugs showed a 3- to 10-fold increase in antiviral activity (0.22, 0.12, and 0.31 µM IC_50_ for **3a, 3b,** and **4a**, respectively) (Figure 3a–d; Table 1). Compared to **1a** (0.79 µM IC_50_), DP prodrug **3a** was 10-fold more active against H7N9 (0.07 µM IC_50_), while prodrugs **3b** (0.62 µM IC_50_) and **4a** (2.02 µM IC_50_) showed comparable or 2.5-fold reduced antiviral activity (Figure 3e–h; Table 1). As already mentioned above, the required concentrations to achieve inhibition of viral replication are non-cytotoxic to the cells, indicating that the observed antiviral activity is not a result of impaired cell viability. Of note, only DP prodrugs **3a** and **3b** reduced H5N1 and H7N9 virus titers to the limit of detection, in contrast to reference **1a** and TP prodrug **4a** (Appendix A).

These findings clearly demonstrate that metabolically pre-activated prodrugs of T-1106 are highly effective in inhibiting the replication of avian flu viruses with known pandemic potential. Importantly, compared to T-705 **1a**, DP prodrug compound **3a** showed improved antiviral efficacy against various seasonal and avian influenza strains and thus might be an interesting candidate for further in vivo studies.

### 3.4. T-1106 Prodrug ***3a*** Acts Synergistically with Oseltamivir against H5N1 HPAIV

Recent epidemic outbreaks of the highly pathogenic avian influenza virus H5N1 in birds, as well as sporadic infections of humans [3], have prompted us to investigate whether our lead prodrug candidate **3a** could also act synergistically with existing therapeutics, such as the neuraminidase inhibitor oseltamivir. In order to define drug concentrations of **3a** and oseltamivir suitable for studying synergistic interactions, we first determined the IC_50_ of oseltamivir against H5N1 replication on MDCKII cells (0.008 µM IC_50_; Appendix A). Next, in addition to the IC_50_ values, we calculated the inhibitory concentrations IC_1_, IC_10_, and IC_90_ for DP prodrug **3a** and oseltamivir, representing a 1%, 10%, or 90% inhibitory effect on viral replication, and used these concentrations for the combination treatment. We infected MDCKII cells with H5N1 at an MOI of 0.01 and therapeutically treated the virus-infected cells with a set of 16 different combinations of both compounds. In addition, we treated the cells with each drug alone (monotherapy). In terms of virus titer reduction, we observed a combinatory effect of the drug combinations that exceeded the efficacy of the respective monotherapies (Appendix A). We then used the open-source web application SynergyFinder Plus to evaluate synergistic interactions of DP prodrug **3a** and oseltamivir. Herein, four different reference models (Bliss, ZIP, HSA, and Loewe) were used to calculate synergy scores that represent the strength of the observed interactions (a high synergy score indicates a strong synergistic effect). Landscape visualization, as well as the calculated synergy scores, revealed a strong synergistic effect of **3a** when combined with oseltamivir, particularly at the two highest concentrations (IC_50_, IC_90_) of both drugs (Figure 4). We further observed positive overall synergy scores independently of the reference model used, indicating, on average, a robust synergistic mode of action. Importantly, no antagonistic effects were detected (Figure 4).

Overall, these data show that our lead DP prodrug compound **3a** acts synergistically with oseltamivir against H5N1 avian influenza virus and thus could possibly be included in combination therapy in the context of pandemic preparedness.

## 4. Discussion

Current influenza treatment options licensed for clinical use are limited and include neuraminidase inhibitors (NAIs), such as oseltamivir, zanamivir, and peramivir; or drugs targeting the viral polymerase (viral polymerase inhibitors, VPIs), including baloxavir or the nucleoside analogue T-705 (Favipiravir) [4]. NAIs inhibit the function of the neuraminidase enzyme and thereby prevent the release of infectious virus particles from the cell [39]. Despite the emergence of drug-resistant viruses, particularly in immunocompromised patients [40,41,42,43], a rather short treatment window of the first two days after the onset of illness [44,45], as well as the uncertain benefit of treating severe influenza cases [6], NAIs are still recommended by various health-care institutions worldwide as the first-line antiviral therapy for influenza infections. VPIs, on the other hand, act on the viral RdRP by various means, ultimately shutting down viral replication and blocking the production of virus progeny. Baloxavir, the only VPI approved in many countries for clinical use, inhibits the RdRP subunit PA (polymerase acidic protein) of various IAV subtypes, including H5N1 and H7N9 HPAIV, with pico- to nanomolar activity [46]. However, shortly after licensing, amino acid substitutions at position 38 in the PA protein (I38T/M/F) that decreased susceptibility to the drug were repeatedly observed in both H1N1 and H3N2 seasonal influenza viruses, suggesting a low barrier of resistance [8,9,47]. Favipiravir, alias T-705, is currently only licensed in Japan to treat IAV resistant to any other class of drug as part of pandemic preparedness. The mode of action was shown to be lethal mutagenesis and/or chain termination after consecutive incorporation, leading to the extinction of infectious virus particles [22,30]. T-705 needs to be converted into the active form by cellular enzymes; hence, this bottleneck in T-705 activation may lead to poor therapeutic efficacy, even at high treatment doses [26]. The therapeutic efficacy of T-705 is further decreased by its chemical instability once it is metabolized into the riboside form [27]. In this study, we aimed to overcome these limitations by chemically synthesizing nucleosides, di- and triphosphate prodrugs of the stabilized T-705 analogue T-1105 (termed T-1106 pronucleotides) that are able to release the active antiviral compounds directly into the infected cell. This was achieved by applying the Di- and Tri*PPP*ro-concept [31]. In previous studies, the chemical stability of the prodrugs was confirmed in either phosphate buffer at neutral pH or when incubated with cell extracts or blood serum [29,48]. Additionally, γ-modified-Tri*PPP*ro-prodrugs were synthesized to achieve a higher selectivity toward viral polymerases. Furthermore, we introduced various chemical modifications (e.g., removal of the 3′-hydroxyl group or addition of a methyl group at the 2′-position of the sugar molecule) to enhance antiviral activity or to decrease the teratogenic/embryotoxic potential by deferring from lethal mutagenesis as the primary mechanism of action. Surprisingly, the majority of the introduced alterations abrogated antiviral activity against a human H3N2 IAV isolate, suggesting that they are no longer tolerated by the RdRP as alternative substrates for viral replication. In fact, we only observed the antiviral activity of T-705 (**1a**) or its de-fluoro analogue T-1105 (**1b**), the ribosylated form of T-1105 (T-1106; **2a**), or three different DP and TP prodrugs (**3a**, **3b**, and **4a**). These prodrugs differ only in the length and size of the lipophilic mask but without any further chemical modifications at the sugar moiety. Improved antiviral efficacy of the DP and TP prodrugs against another H3N2 isolate was also shown by Huchting et al. in 2018 [29]. Furthermore, in this particular study, it was also demonstrated that bypassing cellular T-705 activation pathways is a viable strategy to retain T-705 activity even in the absence of key cellular enzymes involved in the T-705 metabolization [29]. Extending on these findings, here, we also show that DP and TP compounds **3a**, **3b**, and **4a** potently block the replication of a more recent H1N1 IAV strain that was isolated during the 2009 swine flu pandemic. Most importantly, in the present study, we aimed to identify T-705/T-1105 derivates that possess broad-range antiviral activity against various IAV subtypes, including H5N1 and H7N9 HPAIV, which according to the WHO, bear pandemic potential. The efficacy of T-705 against these highly pathogenic viruses has already been demonstrated in vitro and in vivo [49,50,51]. Here, we show that DP and TP prodrugs **3a**, **3b**, and **4a** are highly efficient in inhibiting the replication of both H5N1 and H7N9 HPAIV at low micromolar concentrations. However, we observed a clear impact of the number of phosphate groups attached to the sugar moiety: both DP prodrugs **3a** and **3b** showed 5- to 10-fold increased activity against all viruses tested, compared to the TP compound **4a** and the original T-705 **1a** drug, which is consistent with previous reports [29]. The reduced efficacy of **4a** might be best explained by impaired uptake into the cells due to the additional negatively charged phosphate atom; however, this is yet to be confirmed in future studies. Of note, while T-705 did not affect cell viability at high treatment doses, prodrugs **3a**, **3b**, and **4a** caused considerable cell cytotoxicity starting at 30 µM, again in good agreement with the published literature [29]. Nevertheless, a reduction in viral titers to the limit of detection was observed at concentrations where no major adverse effects on cell viability could be detected. Future work needs to further address potential cytotoxic properties and side effects of the DP and TP prodrugs both in vitro and in vivo.

Prolonged treatment with any therapeutic as well as sub-optimal treatment doses, particularly in situations of compromised immune status such as immunosuppression or pregnancy, may lead to the in-host emergence of viral quasi-species with altered susceptibility to the drug used. This has been repeatedly seen in recent years for both NAIs and VPIs [52,53]. In this regard, it is important to achieve maximal virus inhibition at a safe dose to reduce the likelihood of resistant escape variants evolving under drug-selection pressure. While IC_50_ values are often used to compare the efficacy of a given set of drugs, they do not provide information on residual virus replication in the presence of the treatment. Here, we show that only treatment with the DP and TP prodrugs can reduce both seasonal and HPAIV virus titers to the limit of detection, in sharp contrast to T-705. These findings suggest that T-1106-based prodrugs, which can act independently of cellular metabolic bottlenecks, might be superior to T-705 in terms of preventing the emergence of resistant viruses. It should be noted, however, that even though T-705 resistance could be achieved in cell culture [54], so far, no resistance-conferring amino acid substitutions have been detected in animal models [55] or in clinical trials [56]. Nevertheless, due to the restricted usage of T-705 in humans, it is still unclear whether prolonged and widespread use of the drug might lead to the emergence of resistant influenza viruses.

The ongoing COVID-19 pandemic has revived the debate on whether combination therapy using antiviral agents that differ in their mechanism of action is of clinical interest for treating (severe) influenza infections [57,58]. Combining different drugs may have various benefits over classical monotherapy, such as potentiating the overall antiviral effect, increasing the barrier to viral resistance (particularly within immunocompromised patients), and reducing the effective doses for each individual drug, thereby avoiding the possibility of adverse side effects [57]. Numerous previous reports have already demonstrated synergistic interactions of T-705 with other licensed anti-influenza drugs, such as oseltamivir or baloxavir, both in vitro as well as in animal models and in humans [56,59,60,61,62]. Recent epidemic outbreaks of H5N1 HPAIV in wild birds and sporadic spillovers to mammals, including humans, particularly in the US and in South America [3], have prompted us to investigate potential drug synergies of our lead T-1106 DP prodrug candidate **3a** with oseltamivir against H5N1 HPAIV. Here, we demonstrate that combination treatment effectively reduces H5N1 viral replication to a level that clearly exceeds the effect of the respective monotherapies, confirming a synergistic mode of action. Even though our findings provide the first evidence that T-1106-derived prodrugs may be suited for combinational therapy with oseltamivir, we acknowledge that further in-depth studies using other IAV subtypes as well as other NAIs are required to confirm broad synergistic activity. In addition, possible negative drug-drug interaction effects of the combination treatment need to be evaluated in an appropriate animal model, along with confirming an additive benefit of the drug combination over the respective monotherapies in vivo. Future studies should also address the risk of acquiring resistance against either drug when used as a combination therapy.

In Japan, T-705 use in humans is largely restricted to novel IAVs with resistance to existing drugs. This is mainly due to some early reports on potential teratogenic and embryotoxic properties [25]. Recent studies suggest that Favipiravir might be considerably safer also during pregnancy; however, due to the small number of participants in each study, it is not possible to draw any definite conclusions at this stage [63,64]. We are currently testing both T-705 and our lead prodrug candidates in an allogenic mouse pregnancy model [65] to shed further light on the safety profile of T-705 and its derivates in vivo.

To our knowledge, this is the first study that demonstrates the potent antiviral efficacy of T-705/T-1105-derived prodrugs not only against seasonal but also highly pathogenic avian influenza viruses. Thus, the findings of this study might provide a basis for antiviral therapies against HPAIV, which pose a high pandemic threat.

## 5. Patents

US 10131685, EP 17821806.1.

## Figures and Tables

**Figure 1 pharmaceutics-15-01732-f001:**
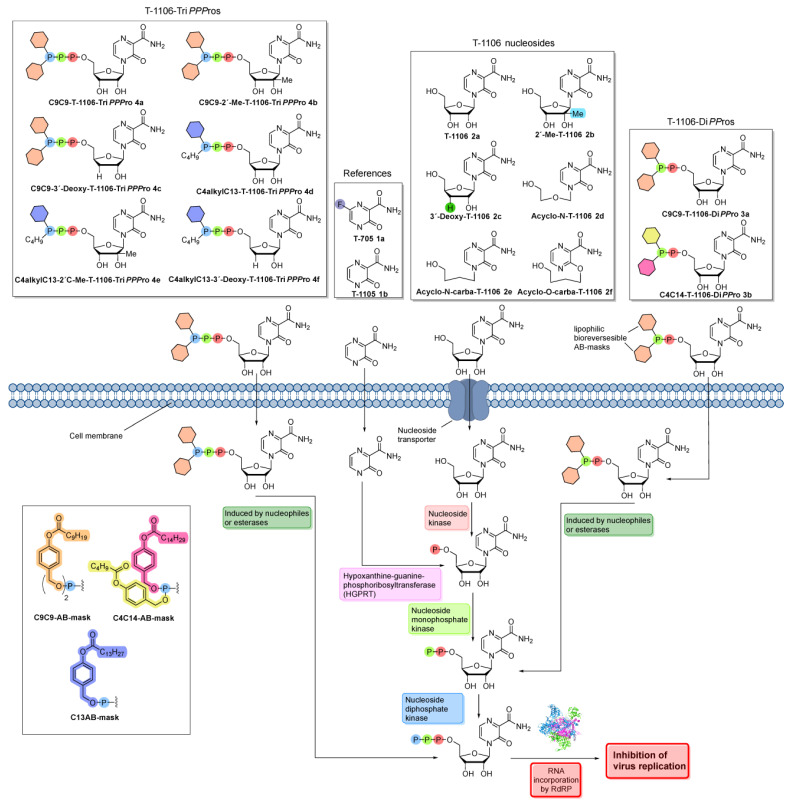
Chemical structures of T-705 **1a** and T-1105 **1b**, T-1106 **2a** and the modified T-1106 nucleosides **2b**–**f**, as well as the T-1106-DiPPro-prodrugs **3a** and **3b** and the T-1106-Tri*PPP*ro-prodrugs **4a**–**4f**. After nucleobases **1a** and **1b** reach the inside of the infected cell via passive diffusion, they are metabolized into their 5′-monophosphates via the humane enzyme HGPRT. The 5′-monophosphates are then further metabolized by human kinases into the antiviral active 5′-triphosphates. The nucleosides **2a**–**f** are transported into the infected cell by nucleoside transporters. Afterward, the nucleosides are metabolized by human kinases into the 5′-triphosphates. The T-1106-Tri*PPP*ro-prodrugs **4a**–**4f** are able to diffuse through the cell membrane due to the lipophilic masking units. Once they are inside the cell, lipophilic masking units are cleaved, and the 5′-triphosphates are released. The T-1106-DiPPro-prodrugs **3a** and **3b** release the 5′-diphosphates instead, which are then metabolized into their 5′-triphosphates.

**Figure 2 pharmaceutics-15-01732-f002:**
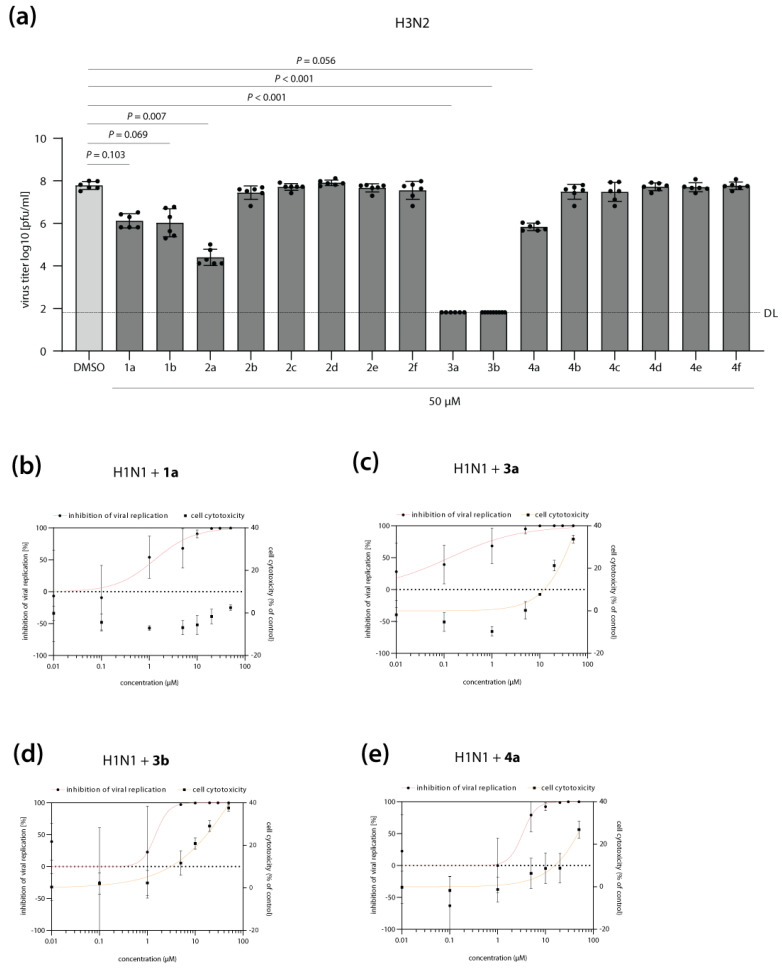
Antiviral activity of T-705 (**1a**) and its derivates against seasonal influenza A virus (IAV) replication. (**a**) MDCKII cells were infected with H3N2 IAV at a multiplicity of infection of 0.01 and subsequently treated with T-705 and its derivates at a final concentration of 50 µM. Cell culture supernatants were collected at 24 h post-infection, and infectious virus titers were determined by plaque assay. DMSO treatment was used as a negative control (0% inhibition). Shown are the average virus titers as log10 plaque forming units per ml (pfu/mL) ± SD of two independent biological replicates, each performed with technical triplicates. Individual data points are shown as well. The detection limit (DL) is indicated with a dashed line. Statistically significant differences were calculated using Kruskal-Wallis test in combination with Dunn’s multiple comparisons tests, and *p* values are displayed. (**b**–**e**, left y-axis) MDCKII cells were infected with H1N1 IAV at a multiplicity of infection of 0.01 and subsequently treated with T-705 **1a** (**b**) or its prodrug derivates **3a** (**c**), **3b** (**d**), and **4a** (**e**) at eight different concentrations (range: 0.01–50 µM). Cell culture supernatants were collected at 24 h post-infection, and infectious virus titers were determined by plaque assay. DMSO treatment was used as a negative control. Shown is the average inhibition of viral replication (%) ± SD by the respective compound, compared to the DMSO control treatment (0% inhibition). Non-linear regression (red curve) was performed to determine the inhibitory concentration 50 (IC_50_) values. (**b**–**e**, right y-axis) MDCK cells were treated with T-705 **1a** (**b**) or its prodrug derivates **3a** (**c**), **3b** (**d**), and **4a** (**e**) at eight different concentrations (range: 0.01–50 µM). At 24 h post-treatment, cell viability was measured using an MTT-based assay according to the manufacturer’s instructions. DMSO treatment was used as a control. The average reduction in cell viability (i.e., cell cytotoxicity, in %) compared to the DMSO control treatment, which was arbitrarily set to 0% cell cytotoxicity. Non-linear regression (orange curve) was performed to determine the cell cytotoxicity 50 (CC_50_) values. Data are derived from two independent biological replicates, each performed in technical triplicates. IC_50_, CC_50_, and SI (selectivity index) values are indicated in Table 1.

**Figure 3 pharmaceutics-15-01732-f003:**
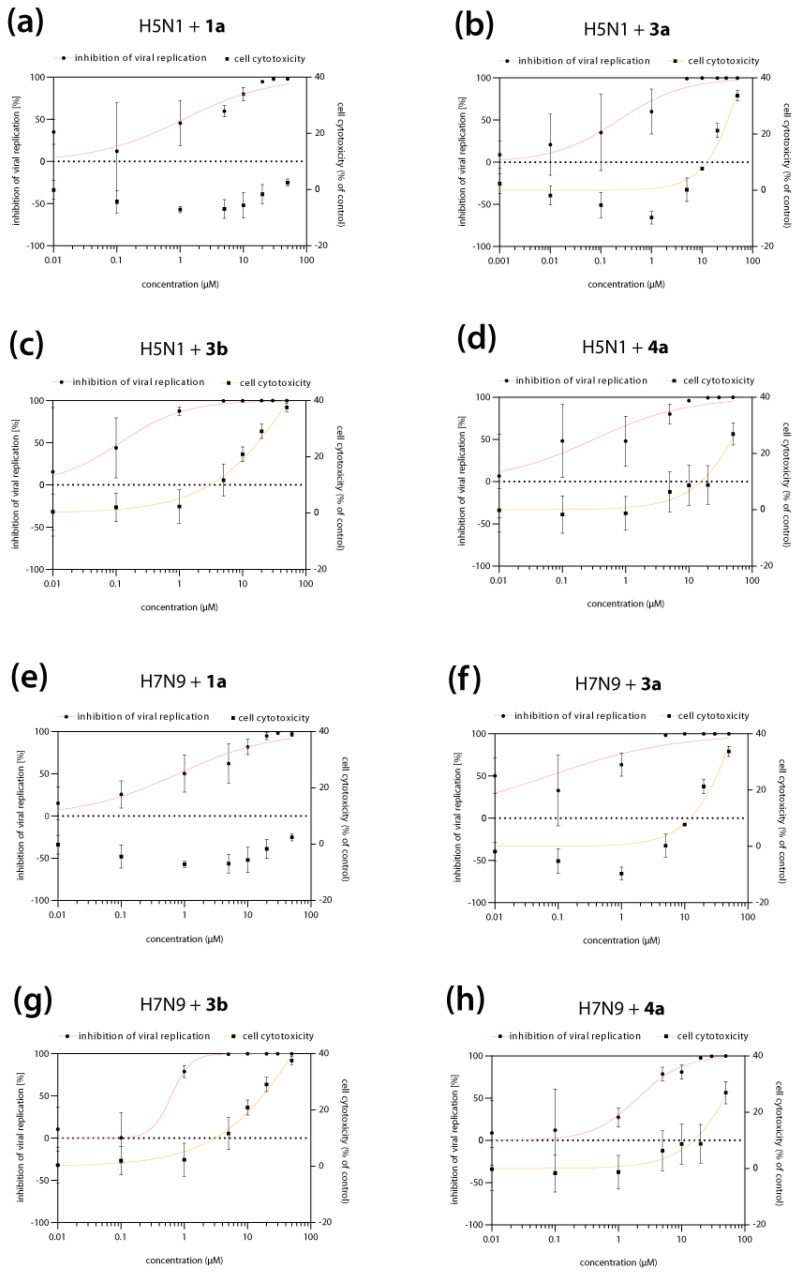
Antiviral activity of T-705 (**1a**) and the T-1106 prodrugs (**3a**, **3b**, and **4a**) against highly pathogenic avian influenza viruses (HPAIV). (**a**–**h**, left y-axis) MDCKII cells were infected with H5N1 (**a**–**d**) or H7N9 (**e**–**h**) HPAIV at a multiplicity of infection of 0.01 and subsequently treated with T-705 **1a** (**a,e**) or its prodrug derivates **3a** (**b**,**f**), **3b** (**c**,**g**), and **4a** (**d**,**h**) at eight different concentrations (range: 0.001–50 µM). Cell culture supernatants were collected at 24 h post-infection, and infectious virus titers were determined by plaque assay. DMSO treatment was used as a negative control. Shown is the average inhibition of viral replication (%) ± SD by the respective compound compared to the DMSO control treatment (0% inhibition). Non-linear regression (red curve) was performed to determine the inhibitory concentration 50 (IC_50_) values. (**a**–**h**, right y-axis) MDCKII cells were treated with T-705 **1a** (**a**,**e**) or its prodrug derivates **3a** (**b**,**f**), **3b** (**c**,**g**), and **4a** (**d**,**h**) at eight different concentrations (range: 0.001–50 µM). At 24 h post-treatment, cell viability was measured using an MTT-based assay according to the manufacturer’s instructions. DMSO treatment was used as a control. The average reduction in cell viability (i.e., cell cytotoxicity, in %) ± SD compared to the DMSO control treatment, which was arbitrarily set to 0% cell cytotoxicity. Non-linear regression (orange curve) was performed to determine the cell cytotoxicity 50 (CC_50_) values. (**a**–**h**) Data are derived from two independent biological replicates, each performed in technical triplicates. IC_50_, CC_50_, and SI (selectivity index) values are indicated in Table 1.

**Figure 4 pharmaceutics-15-01732-f004:**
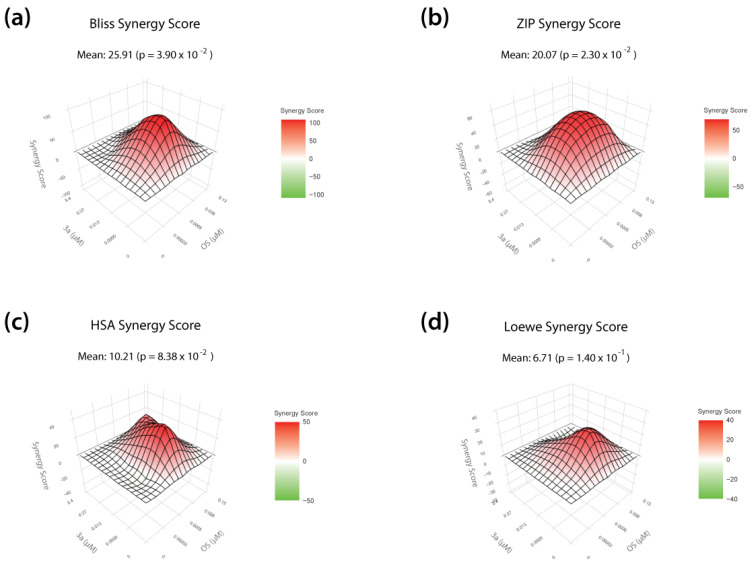
Antiviral activity of T-1106-derived DP prodrug **3a** in combination with the neuraminidase inhibitor oseltamivir (OS) against H5N1 HPAIV replication. (**a**–**d**) MDCKII cells were infected with H5N1 HPAIV at a multiplicity of infection of 0.01 and subsequently treated with combinations of the T-1106 prodrug **3a** and the neuraminidase inhibitor oseltamivir. The tested concentrations are based on the respective IC_1_, IC_10_, IC_50_, and IC_90_ values for each drug. Cell culture supernatants were collected at 24 h post infection and infectious virus titers were determined by plaque assay. DMSO treatment was used as a negative control (0% inhibition). Bliss independence (Bliss; **a**), Zero Interaction Potency (ZIP; **b**), highest single agent (HSA; **c**), and Loewe additivity (Loewe; **d**) reference models were used to calculate and visualize synergistic areas. The shown landscapes are color-coded into synergistic (red) and antagonistic (green) interactions. Data are derived from three independent biological replicates, each performed in technical triplicates.

**Table 1 pharmaceutics-15-01732-t001:** CC_50_ (cell cytotoxicity 50), IC_50_ (inhibitory concentration 50), and SI (selectivity index) values of T-705 **1a** and the T-1106 prodrugs **3a**, **3b**, and **4a**.

ID	MDCK	H1N1		H5N1		H7N9	
	CC_50_ (µM) ^1^	IC_50_ (µM) ^1^	SI	IC_50_ (µM) ^1^	SI	IC_50_ (µM) ^1^	SI
**1a**	>50	1.295 ± 1.40	n.a.	1.138 ± 1.17	n.a.	0.790 ± 0.54	n.a.
**3a**	83.71 ± 11.45	0.130 ± 0.14	643.9	0.224 ± 0.23	373.7	0.071 ± 0.09	1179.0
**3b**	98.70 ± 16.05	1.486 ± 2.36	66.4	0.116 ± 0.013	850.9	0.624 ± 1.04	158.2
**4a**	144.5 ± 106.61	3.318 ± 4.41	43.6	0.314 ± 0.38	460.2	2.020 ± 1.23	71.5

^1^ Data are derived from two independent biological replicates, each performed in technical triplicates. CC_50_ and IC_50_ values are shown as average ± SD.

## Data Availability

Data are contained within the article or Appendix A.

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
