# Peer review of "T-705-Derived Prodrugs Show High Antiviral Efficacies against a Broad Range of Influenza A Viruses with Synergistic Effects When Combined with Oseltamivir"

_pharmaceutics, 2023, doi:10.3390/pharmaceutics15061732_

Round 1

Reviewer 1 Report

The article is well-written and organized. It is not, however, suitable for the Pharmaceutics journal. There is no aspect of formulation and evaluation. The manuscript is suitable for publication in either a biological or a chemistry journal. 

The article is well-written and organized. It is not, however, suitable for the Pharmaceutics journal. There is no aspect of formulation and evaluation. The manuscript is suitable for publication in either a biological or a chemistry journal. 

Author Response

We thank the reviewer for his/her feedback. We submitted our manuscript to the special issue "drugs for antiviral combination therapy". Our study is supported to be within the scope of Pharmaceutics by the other reviewers.

Reviewer 2 Report

This work synthesized T-705-derived prodrugs and demonstrated their high antiviral efficacies against a 2 broad range of influenza A viruses. However, the present results were not so solid.

(1)   Why did the T-705-derived prodrugs show the high antiviral efficacies? The investigation of the mechanism should be provided.

(2)   The structure characterizations of the T-705-derived prodrugs should be provided.

(3)   So many quantitative results of the data, please provide some image results about the antiviral activity.

(4)   The antiviral activity in vivo should be investigated.

Extensive editing of English language required

Author Response

This work synthesized T-705-derived prodrugs and demonstrated their high antiviral efficacies against a broad range of influenza A viruses. However, the present results were not so solid.

We thank the reviewer for his/her feedback.

(1)   Why did the T-705-derived prodrugs show the high antiviral efficacies? The investigation of the mechanism should be provided.

The mechanism of action for T-705 was described before to be lethal mutagenesis and/or chain termination (Huchting et al., J Med Chem, 2018, DOI: 10.1021/acs.jmedchem.8b00617; Kouba et al., Cell Rep., 2023, DOI: 10.1016/j.celrep.2022.111901). The proposed mode-of-action is discussed accordingly (see lines 887-906).

(2)   The structure characterizations of the T-705-derived prodrugs should be provided.

In Figure 1, we show the structures of the nucleosides and the prodrugs. We now also cite a previous study, where further structural information is provided (Huchting et al., J Med Chem, 2018, DOI: 10.1021/acs.jmedchem.8b00617; see lines 585-586).

(3)   So many quantitative results of the data, please provide some image results about the antiviral activity.

We believe that showing quantitative data is the best way to allow thorough data interpretation. However, we understand that a second method more convenient to the eye is requested.  We now included an additional figure showing viral plaques on MDCK cells upon compound treatment (see new Supplementary Figure S2).

(4)   The antiviral activity in vivo should be investigated.

We agree that these studies need to be performed next. However, these studies are out of the scope of the current study.

Reviewer 3 Report

In this manuscript, the derivative library of T705 ribonucleoside analogues was synthesized and their ability to inhibit seasonal and highly pathogenic avian influenza virus in vitro was tested. In addition, the drug synergism of neuraminidase inhibitor oseltamivir and leading DP prodrug derivatives was studied. Overall, this is a well-designed study.

Author Response

We thank the reviewer for his/her positive feedback.

Reviewer 4 Report

This paper reported the discovery of T-705 analogs as the promising antiviral agents targeting a broad range of influenza A viruses (IAV). Especially, T-1106-DP as the prodrug form of T705 analog showed potent inhibition of IAV without overt cytotoxicity. Moreover, T-1106-DP acts synergistically with oseltamivir against H5N1 avian influenza virus and has the potential to be included in the future combination therapy. The topic fits the scope of this journal, and may potentially benefit the future treatment of IAV infections in the context of pandemic preparedness. In general, this manuscript is organized well and the experiments can basically support the conclusion. Even though, key issues are required to be addressed before its publication on Pharmaceutics.

Major points:

1. This study identified the T-705 analogs showing significantly improved antiviral activities against IAV, while the target validation was not performed. The readers are not clear if these compounds still work on the RdRp or the other process during of IAV infection. The authors are required to test the biochemical activity, e.g. binding affinity (to RdRp), of these new analogs, such as T-1106-DP, or use MS spectra to characterize this compound really involving in the lethal mutagenesis and/or chain termination of virus RNA.

2. How about the stability of these prodrug? Is there any stability test the author conducted? This is another big concern considering the di- or tri-phosphate structures of these compounds.

3. In Table 1, the cytotoxicity of 1a (T-705) is required to be conducted or referenced, as it is important to the comparison of new analogs with the parent compound.

4. In Table 1, the standard deviations (SD) of these values are required.  

Minor points:

1. In the introduction section, a Figure illustration about the research background as well as the current design strategy in brief is suggested. Some information may be directly extracted from the current Figure 1 in the Results section.

2. The technical repeats are required to be mentioned in the method section of antiviral assay, or to be footnoted under Table 1 just as that in Figure 3 legend.

English writing is good.

Author Response

We thank the reviewer for his/her response. 

Major points:
1. This study identified the T-705 analogs showing significantly improved antiviral activities against IAV, while the target validation was not performed. The readers are not clear if these compounds still work on the RdRp or the other process during of IAV infection. The authors are required to test the biochemical activity, e.g. binding affinity (to RdRp), of these new analogs, such as T-1106-DP, or use MS spectra to characterize this compound really involving in the lethal mutagenesis and/or chain termination of virus RNA.

Yes, the reviewer is right that these mechanistic studies need to be performed in detail in future experiments. We have shown before that these prodrug derivates still act on the RdRP (Huchting et al., J Med Chem, 2018, DOI: 10.1021/acs.jmedchem.8b00617). The focus of our current study was to identify compounds that enter the cells and possess broad-spectrum antiviral activity. In this study, we describe hit candidates that are able to reduce the replication of two key seasonal human influenza A viruses (H1N1, H3N2) as well as of two avian influenza A viruses that are of pandemic concern (H5N1, H7N9).

2. How about the stability of these prodrugs? Is there any stability test the author conducted? This is another big concern considering the di- or tri-phosphate structures of these compounds.

The prodrugs are stabil in phosphate buffer, pH 7.3, but are rapidly converted in the presence of cellular esterases into the de-masked antiviral active compound, which is intended as such.  Furthermore, prodrugs incubated in blood serum or with cell extracts also proved to be very stabil over hours (Huchting et al., J Med Chem, 2018, DOI: 10.1021/acs.jmedchem.8b00617; Zhao et al., Angew Chem Int Ed, 2020, DOI: 10.1002/anie.202003073). As long as the di- or the triphosphate moiety is masked at the γ- or β-phosphate, the di- or triphosphate group is stabil. However, after delivery of the corresponding free di- or triphosphates, a dephosphorylation may occur which is also nucleoside dependent. We now mention these features in the discussion section, respectively (see lines 899-901).

3. In Table 1, the cytotoxicity of 1a (T-705) is required to be conducted or referenced, as it is important to the comparison of new analogs with the parent compound.

We agree and have now included this information (see revised Table 1, lines 797-800). As shown in Figure 2b, Figure 3a and Figure 3e (right y axes), we could not detect any cytotoxicity of compound 1a (T-705) up to 50 µM. This is in agreement with the published literature, where even concentrations up to 250 µM did not cause cytotoxicity on MDCK cells (Huchting et al., J Med Chem, 2018, DOI: 10.1021/acs.jmedchem.8b00617; see lines 931-933).

4. In Table 1, the standard deviations (SD) of these values are required.

Yes, we have now included the standard deviations for the respective CC50 and IC50 values (see revised Table 1, lines 797-800). 

Minor points:
1. In the introduction section, a Figure illustration about the research background as well as the current design strategy in brief is suggested. Some information may be directly extracted from the current Figure 1 in the Results section.

According to the journal guidelines, we are not allowed to include a figure in the introduction section of the manuscript. However, we believe that all the information on T-705 history of development, mechanism of action and current drawbacks for clinical use, including citation of the relevant and most recent literature, is sufficiently described in the introduction section.

2. The technical repeats are required to be mentioned in the method section of antiviral assay, or to be footnoted under Table 1 just as that in Figure 3 legend.

The IC50 values presented in revised Table 1 are directly derived from the data shown in Figures 2b-e and Figure 3 for which the number of technical and biological replicates is indicated in the respective figure legends. However, as requested, we have now included this information also in the revised Table 1 as a footnote (see lines 799-800).

Round 2

Reviewer 2 Report

It might be acceptable.

Author Response

We thank the reviewer for supporting the publication of our revised manuscript in Pharmaceutics.

Reviewer 4 Report

Accept

Author Response

(The authors gave the same response as above.)
